# Hyperparameter Tuning is All You Need for LISTA

**Xiaohan Chen**[1*]   **Jialin Liu**[2*]   **Zhangyang Wang**[1]   **Wotao Yin**[2]

[1]University of Texas at Austin    [2]Alibaba US, Damo Academy
{xiaohan.chen, atlaswang}@utexas.edu
{jialin.liu, wotao.yin}@alibaba-inc.com

## Abstract

Learned Iterative Shrinkage-Thresholding Algorithm (LISTA) introduces the concept of unrolling an iterative algorithm and training it like a neural network. It has had great success on sparse recovery. In this paper, we show that adding momentum to intermediate variables in the LISTA network achieves a better convergence rate and, in particular, the network with instance-optimal parameters is superlinearly convergent. Moreover, our new theoretical results lead to a practical approach of automatically and adaptively calculating the parameters of a LISTA network layer based on its previous layers. Perhaps most surprisingly, such an adaptive-parameter procedure reduces the training of LISTA to tuning **only three hyperparameters** from data: a new record set in the context of the recent advances on trimming down LISTA complexity. We call this new ultra-light weight network *HyperLISTA*. Compared to state-of-the-art LISTA models, HyperLISTA achieves almost the same performance on seen data distributions and performs better when tested on unseen distributions (specifically, those with different sparsity levels and nonzero magnitudes). Code is available: `https://github.com/VITA-Group/HyperLISTA`.

## 1 Introduction

In this paper, we study the sparse linear inverse problem, where we strive to recover an unknown sparse vector $x^* \in \mathbb{R}^n$ from its noisy linear measurement $b$ generated from

$$b = Ax^* + \varepsilon, \tag{1}$$

where $b \in \mathbb{R}^m$ is the measurement that we observe, $A \in \mathbb{R}^{m \times n}$ is the *dictionary*, $x^* \in \mathbb{R}^n$ is the unknown ground truth that we aim to recover, and $\varepsilon \in \mathbb{R}^m$ is additive Gaussian white noise. For simplicity, each column of $A$, is normalized to have unit $\ell_2$ norm. Typically, we have much fewer rows than columns in the dictionary $A$, i.e., $m \ll n$. Therefore, Equation (1) is an under-determined system. The sparse inverse problem, also known as sparse coding, plays essential roles in a wide range of applications including feature selection, signal reconstruction and pattern recognition.

Sparse linear inverse problems are well studied in the literature of optimization. For example, it can be formulated into LASSO [29] and solved by many optimization algorithms [9, 3]. These solutions explicitly consider and incorporate the sparsity prior and usually exploit iterative routines. Deep-learning-based approaches are proposed for empirically solving inverse problems recently [25], which produce black-box models that are trained with data in an end-to-end way, while somehow ignoring the sparsity prior. Comparing the two streams, the former type, i.e., the classic optimization methods, takes hundreds to thousands of iterations to generate accurate solutions, while black-box deep learning models, if properly trained, can achieve similar accuracy in tens of layers. However, classic methods come with data-agnostic convergence (rate) guarantees under suitable conditions,

35th Conference on Neural Information Processing Systems (NeurIPS 2021).

---

[*]The first two authors made equal contributions.

whereas deep learning methods only empirically work for instances similar to the training data, lacking theoretical guarantees and interpretability. More discussions are found in [6].

*Unrolling* is an uprising approach that bridges the two streams [22, 21]. By converting each iteration of a classic iterative algorithm into one layer of a neural network, one can *unroll* and truncate a classic optimization method into a feed-forward neural network, with a finite number of layers. Relevant parameters (e.g., the dictionary, step sizes and thresholds) in the original algorithm are transformed into learnable weights, that are trained on data. [15] pioneered the unrolling scheme for solving sparse coding and achieved great empirical success. By unrolling the Iterative Shrinkage-Thresholding Algorithm (ISTA), the authors empirically demonstrated up to two magnitudes of acceleration of the learned ISTA (LISTA) compared to the original ISTA. The similar unrolling idea was later extended to numerous optimization problems and algorithms [28, 30, 26, 13, 36, 2, 16, 8, 5].

## 1.1 Related Works

The balance between empirical performance and interpretability of unrolling motivates a line of theoretical efforts to explain their acceleration success. [23] attempted to understand LISTA by re-factorizing the Gram matrix of the dictionary $A$ in (1) and proved that such re-parameterization converges sublinearly and empirically showed that it achieved similar acceleration gain to LISTA. [33] investigated unrolling iterative hard thresholding (IHT) and proved the existence of a matrix transformation that can improve the restricted isometry property constant of the original dictionary. Although it is difficult to search for the optimal transformation using classic optimization techniques, the authors of [33] argued that the data-driven training process is able to learn that transformation from data, and hence achieving the acceleration effect in practice.

The work [7] extended the weight coupling necessary condition in [33] to LISTA. The authors for the first time proved the existence of a learnable parameter set that guarantees linear convergence of LISTA. Later in [18], they further showed that the weight matrix in [7] can be analytically derived from the dictionary $A$ and free from learning, reducing the learnable parameter set to only tens of scalars (layer-wise step sizes and thresholds). The resulting algorithm, called Analytic LISTA (ALISTA), sets one milestone in simplifying LISTA. [31] extends ALISTA by introducing gain and overshoot gating mechanisms and integrate them into unrolling. [37] proposed error-based thresholding (EBT), which leverages a function of the layer-wise reconstruction error to suggest an appropriate threshold value for each observation on each layer. Another interesting work [4] uses a black-box LSTM model that takes the historical residual and update quantity as input and generates layerwise step sizes and thresholds in ALISTA. Their proposed model, called Neurally Augmented ALISTA (NA-ALISTA), performed well in large-scale problems.

## 1.2 Our Contributions

Our research question is along the line of [18, 37, 4]: *can we improve the LISTA parameterization, by further disentangling the learnable parameters from the observable terms (e.g., reconstruction errors, intermediate outputs)*? Our aim is to create more light-weight, albeit more robust LISTA models with "baked in" adaptivity to testing samples from unseen distributions. The goal should be achieved without sacrificing our previously attained benefits: empirical performance, and convergence (rate) guarantees – if not improving them further.

We present an affirmative answer to the above question by learning instance-optimal parameters with a new ALISTA parameterization. Under consideration, we prove that by augmenting ALISTA with momentum, the new unrolled network can achieve a better convergence rate over ALISTA. Second, we find that once the above LISTA network is further enabled with instance-optimal parameters, then a superlinear convergence can be attained. Lastly, we show that those instance-optimal parameters of each layer can be generated in closed forms, consequently reducing ALISTA to only learning **three instance- and layer-invariant hyperparameters**.

Our new ultra-light weight network is termed as *HyperLISTA*. Compared to ALISTA, the new parameterization of HyperLISTA facilitates new backpropagation-free "training" options (e.g., bayesian optimization), and can unroll to more iterations in testing than training thanks to its layer-wise decomposed form. When comparing to state-of-the-art LISTA variants in experiments, HyperLISTA achieves the same strong performance on seen data distributions, and performs better when tested on unseen distributions (e.g., with different sparsity levels and nonzero magnitudes). In the supple-

mentary materials, we also show that the superlinear convergence can be observed in HyperLISTA experiments, when our automatic parameter tuning is performed per instance.

## 2 Assumptions and Preliminaries

**Mathematical Notations:** We represent a matrix as a capital bold letter $\boldsymbol{W}$, $\mathrm{diag}(\boldsymbol{W})$ means the diagonal elements of matrix $\boldsymbol{W}$ and the pseudoinverse of a matrix is denoted by $\boldsymbol{W}^+$. A vector is represented as a lowercase bold letter $\boldsymbol{x}$ and its entries are represented as lowercase regular letters with lower indices $\boldsymbol{x} = [x_1, x_2, \cdots, x_3]^T$. Upper indices represent iterations or layers. Moreover, $\boldsymbol{0}, \boldsymbol{1}$ represent a vector with all zeros or all ones, respectively. "$\mathrm{supp}(\boldsymbol{x})$" denotes the support, the indices of all nonzeros in vector $\boldsymbol{x}$.

**ISTA Algorithm:** With an $\ell_1$ penalty term, the minimizer of LASSO, $\mathrm{argmin}_{\boldsymbol{x}} \frac{1}{2}\|\boldsymbol{Ax}-\boldsymbol{b}\|_2^2+\lambda\|\boldsymbol{x}\|_1$, provides an estimate of the solution of the sparse recovery problem (1). A popular algorithm to solve LASSO is Iterative Shrinkage-Thresholding Algorithm (ISTA):

$$\boldsymbol{x}^{(k+1)} = \eta_{\lambda/L}\Big(\boldsymbol{x}^{(k)} + \frac{1}{L}\boldsymbol{A}^T(\boldsymbol{b} - \boldsymbol{Ax}^{(k)})\Big), \quad k = 0, 1, 2, \dots$$

where $\eta_{\lambda/L}$ is the element-wise soft-thresholding function $\eta_\theta(\boldsymbol{x}) = \mathrm{sign}(\boldsymbol{x})\max(0, |\boldsymbol{x}| - \theta)$ with $\theta = \lambda/L$ and $L$ is usually taken as the largest eigenvalue of $\boldsymbol{A}^T\boldsymbol{A}$.

**LISTA Model:** The original LISTA work [15] parameterizes ISTA as

$$\boldsymbol{x}^{(k+1)} = \eta_{\theta^{(k)}}\Big(\boldsymbol{W}_1^{(k)}\boldsymbol{x}^{(k)} + \boldsymbol{W}_2^{(k)}\boldsymbol{b}\Big), \quad k = 0, 1, 2, \dots \tag{2}$$

where $\Theta = \{\boldsymbol{W}_1^{(k)}, \boldsymbol{W}_2^{(k)}, \theta^{(k)}\}$ are learnable parameters to train from data. In the training process, iterations formulated in (2) are unrolled and truncated to $K$ steps and the parameters are trained by minimizing the following loss function defined over the distribution of training samples $D_{\mathrm{tr}}$:

$$\underset{\Theta}{\mathrm{minimize}} \, \mathbb{E}_{(\boldsymbol{x}^*, \boldsymbol{b}) \sim D_{\mathrm{tr}}} \Big\| \boldsymbol{x}^{(K)}(\Theta, \boldsymbol{b}, \boldsymbol{x}^{(0)}) - \boldsymbol{x}^* \Big\|_2^2. \tag{3}$$

Here $\boldsymbol{x}^{(K)}(\Theta, \boldsymbol{b}, \boldsymbol{x}^{(0)})$, the output of the LISTA model, is a function with input $\Theta$, $\boldsymbol{b}$ and $\boldsymbol{x}^{(0)}$. In practice, we usually set $\boldsymbol{x}^{(0)} = \boldsymbol{0}$. We will use $\boldsymbol{x}^{(K)}$ to refer to $\boldsymbol{x}^{(K)}(\Theta, \boldsymbol{b}, \boldsymbol{x}^{(0)})$ for brevity in the remainder of this paper. $\boldsymbol{x}^*$ denotes the underlying ground truth defined in (1). Although it cannot be accessed in iterative algorithm (2), it can be used as the label in supervised training (3).

**ALISTA Model:** ALISTA [18] advances LISTA with a lighter and more interpretable parameterization, where the learnable parameters are $\Theta = \{\gamma^{(k)}, \theta^{(k)}\}$ and the iterations are formulated as

$$\boldsymbol{x}^{(k+1)} = \eta_{\theta^{(k)}}^{p^{(k)}}\Big(\boldsymbol{x}^{(k)} + \gamma^{(k)}\boldsymbol{W}^T(\boldsymbol{b} - \boldsymbol{Ax}^{(k)})\Big), \tag{4}$$

where $\boldsymbol{W}$ is calculated by solving the following optimization problem:

$$\boldsymbol{W} \in \underset{\boldsymbol{W} \in \mathbb{R}^{m \times n}}{\mathrm{argmin}} \big\| \boldsymbol{W}^T\boldsymbol{A} \big\|_F^2, \quad \text{s.t. } \mathrm{diag}(\boldsymbol{W}^T\boldsymbol{A}) = \boldsymbol{1}. \tag{5}$$

The thresholding operator is improved with support selection. At layer $k$, a certain percentage of entries with largest magnitudes are trusted as "true support" and will not be passed through thresholding. Specifically, the $i$-th element of $\eta_{\theta^{(k)}}^{p^{(k)}}(\boldsymbol{v})$ is defined as

$$(\eta_{\theta^{(k)}}^{p^{(k)}}(\boldsymbol{v}))_i = \begin{cases} v_i & : v_i > \theta^{(k)}, & i \in S^{p^{(k)}}(\boldsymbol{v}), \\ v_i - \theta^{(k)} & : v_i > \theta^{(k)}, & i \notin S^{p^{(k)}}(\boldsymbol{v}), \\ 0 & : -\theta^{(k)} \le v_i \le \theta^{(k)} \\ v_i + \theta^{(k)} & : v_i < -\theta^{(k)}, & i \notin S^{p^{(k)}}(\boldsymbol{v}), \\ v_i & : v_i < -\theta^{(k)}, & i \in S^{p^{(k)}}(\boldsymbol{v}), \end{cases} \tag{6}$$

where $S^{p^{(k)}}(\boldsymbol{v})$ includes the elements with the largest $p^{(k)}$ magnitudes in vector $\boldsymbol{v} \in \mathbb{R}^n$:

$$S^{p^{(k)}}(\boldsymbol{v}) = \Big\{ i_1, i_2, \cdots, i_{p^{(k)}} \Big| |v_{i_1}| \ge |v_{i_2}| \ge \cdots |v_{i_{p^{(k)}}}| \cdots \ge |v_{i_n}| \Big\}. \tag{7}$$

With $p^{(k)} = 0$, the operator reduces to the soft-thresholding and, with $p^{(k)} = n$, the operator reduces to the hard-thresholding. Thus, operator $\eta^{p^{(k)}}_{\theta^{(k)}}(\boldsymbol{v})$ is actually a balance between soft- and hard-thresholding. When $k$ is small, $p^{(k)}$ is usually chosen as a small fraction of $n$ and the operator tends to not trust the signal, and vice versa. In ALISTA, $p^{(k)}$ is proportional to the index $k$, capped by a maximal value, i.e., $p^{(k)} = \min(p \cdot k, p_{\max})$, where $p$ and $p_{\max}$ are two hyperparameters that will be tuned manually. Later in this work, we will consider $p^{(k)}$ as a parameter that could be adaptively calculated by the information from previous iterations (layers).

**More Notations and Assumptions:** Throughout this paper, we refer to "parameters" as the learnable weights in the models, such as the thresholds $\theta^{(k)}$ and the step sizes $\gamma^{(k)}$ that are defined in the ALISTA model, which are usually trained by back-propagation. In contrast, we refer "hyperparameters" as those constants that are often ad-hoc pre-defined to calculate parameters. For example, $\theta^{(k)} = c_1 \|\boldsymbol{A}^+(\boldsymbol{A}\boldsymbol{x}^{(k)} - \boldsymbol{b})\|_1$, where $\theta^{(k)}$ is a parameter and $c_1$ is a hyperparameter.

**Assumption 1** (Basic assumptions). *The signal $\boldsymbol{x}^*$ and noise $\varepsilon$ are sampled from the following set:*

$$\mathcal{X}(B, \underline{B}, s, \sigma) \triangleq \left\{ (\boldsymbol{x}^*, \varepsilon) \Big| \|\varepsilon\|_1 \leq \sigma, \ \|\boldsymbol{x}^*\|_0 \leq s, \ 0 < \underline{B} \leq |\boldsymbol{x}^*_i| \leq B, \ \forall i \in \mathrm{supp}(\boldsymbol{x}^*) \right\}. \quad (8)$$

*In other words, $\boldsymbol{x}^*$ is bounded and sparse. $\sigma > 0$ is the noise level.*

Compared with the Assumption 2 in [7], the assumption of "$|\boldsymbol{x}^*_i| \geq \underline{B} > 0$" for non-zero elements in $x^*$ is slightly stronger. But we will justify (after Theorem 1) that uniformly under Assumption 1 in this paper, our method also achieves a better convergence rate than state-of-the-arts.

## 3 Methodologies and Theories

In this section, we extend the ALISTA (4) in three aspects. First, we introduce a better method to calculate matrix $\boldsymbol{W}$ in formula (5). Second, we augment the x-update formula (4) by adding a momentum term. Finally, we propose an adaptive approach to obtain the parameters $p^{(k)}, \theta^{(k)}, \gamma^{(k)}$.

### 3.1 Preparation: Symmetric Jacobian of Gradients

Before improving the formula (5), we first briefly explain its main concepts and shortcomings. Mutual coherence is a key concept in compressive sensing[11, 10]. For matrix $\boldsymbol{A}$, it is defined as $\max_{i \neq j} |(\boldsymbol{A}^T \boldsymbol{A})_{i,j}|$ where $(\boldsymbol{A}^T \boldsymbol{A})_{i,i} = 1$ since each column of $\boldsymbol{A}$ is assumed with unit $\ell_2$ norm. A matrix with low mutual coherence satisfies that $\boldsymbol{A}^T \boldsymbol{A}$ approximates the identity matrix, thus implying a recovery of high probability. In ALISTA [18], the authors extended this concept to a relaxed bound between matrix $\boldsymbol{A}$ and another matrix $\boldsymbol{W}$: $\|\boldsymbol{W}^T \boldsymbol{A}\|_F$ (described in (5)). By minimizing this bound, they obtained a good matrix $\boldsymbol{W}$ that can be plugged into the LISTA framework.

One clear limitation of ALISTA is that, with $\boldsymbol{W}$ obtained by (5), the model is only able to converge to $\boldsymbol{x}^*$ but not the LASSO minimizer [2]. Consequently, one has to train ALISTA with known $\boldsymbol{x}^*$ in a supervised way (3). This is partially due to the fact that the update direction $\boldsymbol{W}^T(\boldsymbol{A}\boldsymbol{x} - \boldsymbol{b})$ in ALISTA is not aligned with the gradient of the $\ell_2$ term in the LASSO objective: $\nabla_{\boldsymbol{x}} \frac{1}{2} \|\boldsymbol{A}\boldsymbol{x} - \boldsymbol{b}\|_2^2 = \boldsymbol{A}^T(\boldsymbol{A}\boldsymbol{x} - \boldsymbol{b})$. This limitation motivates us to adopt a new symmetric Jacobian parameterization so that $\boldsymbol{W}^T \boldsymbol{A}$ is symmetric and the coherence between $\boldsymbol{A}$ and $\boldsymbol{W}$ is minimized.

Inspired by [1], we define $\boldsymbol{W} = (\boldsymbol{G}^T \boldsymbol{G})\boldsymbol{A}$ (the matrix $\boldsymbol{G} \in \mathbb{R}^{m \times m}$ is named as the Gram matrix), and get the following problem instead of (5):

$$\min_{\boldsymbol{G}} \|\boldsymbol{A}^T \boldsymbol{G}^T \boldsymbol{G} \boldsymbol{A} - \boldsymbol{I}\|_F^2, \quad \text{s.t. } \mathrm{diag}(\boldsymbol{A}^T \boldsymbol{G}^T \boldsymbol{G} \boldsymbol{A}) = \boldsymbol{1}. \quad (9)$$

However, the constraint in the above problem (9) is hard to handle. Thus, we introduce an extra matrix $\boldsymbol{D} = \boldsymbol{G}\boldsymbol{A} \in \mathbb{R}^{m \times n}$ and use the following method instead to calculate dictionary:

$$\min_{\boldsymbol{G}, \boldsymbol{D}} \|\boldsymbol{D}^T \boldsymbol{D} - \boldsymbol{I}\|_F^2 + \frac{1}{\alpha} \|\boldsymbol{D} - \boldsymbol{G}\boldsymbol{A}\|_F^2, \quad \text{s.t. } \mathrm{diag}(\boldsymbol{D}^T \boldsymbol{D}) = \boldsymbol{1}. \quad (10)$$

With proper $\alpha > 0$, the solution of (10) approximates (9) well. We use an adaptive rule to choose $\alpha$ and adopt a variant of the algorithm (described in details in the supplement) proposed in [19] to solve (10). Such formula leads to a symmetric matrix $\boldsymbol{W}^T \boldsymbol{A} = (\boldsymbol{G}\boldsymbol{A})^T(\boldsymbol{G}\boldsymbol{A})$. With this symmetry, the vector $\boldsymbol{W}^T(\boldsymbol{A}\boldsymbol{x} - \boldsymbol{b})$ is actually the gradient of function $\frac{1}{2}\|\boldsymbol{G}(\boldsymbol{A}\boldsymbol{x} - \boldsymbol{b})\|_2^2$ with respect to $\boldsymbol{x}$.

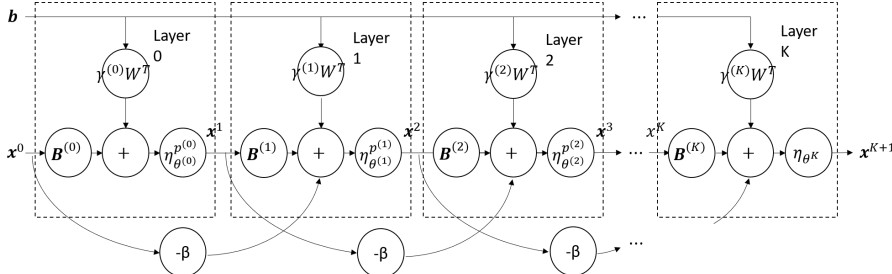

Figure 1: ALISTA with momentum as formulated in (12), with $\boldsymbol{B}^{(0)} = \boldsymbol{I} - \gamma^{(0)}\boldsymbol{W}^T\boldsymbol{A}$, $\boldsymbol{B}^{(k)} = \left(1+\beta^{(k)}\right)\boldsymbol{I} - \gamma^{(k)}\boldsymbol{W}^T\boldsymbol{A}$ ($k \geq 1$). The momentum creates extra skip connections at the bottom.

One may train model (4) using methods in [2] with the following loss function without knowing the ground truth: $F(\boldsymbol{x}) = \frac{1}{2}\|\boldsymbol{G}(\boldsymbol{A}\boldsymbol{x} - \boldsymbol{b})\|_2^2 + \lambda\|\boldsymbol{x}\|_1$. On the other hand, it usually holds that

$$\|\boldsymbol{W}^T\boldsymbol{A} - \boldsymbol{I}\|_F \approx \|(\boldsymbol{G}\boldsymbol{A})^T(\boldsymbol{G}\boldsymbol{A}) - \boldsymbol{I}\|_F, \tag{11}$$

where $\boldsymbol{W}$ is calculated by (5) and $\boldsymbol{G}$ is calculated by (10). We validate (11) by numerical results in Section 4.1. Conclusion (11) demonstrates that the mutual coherence between $\boldsymbol{A}$ and $(\boldsymbol{G}^T\boldsymbol{G})\boldsymbol{A}$ is similar to that between $\boldsymbol{A}$ and $\boldsymbol{W}$ obtained by (5). Although we limit the matrix $\boldsymbol{W}^T\boldsymbol{A}$ to be symmetric, (10) hardly degrades any performance of the ALISTA framework compared to (5).

### 3.2 ALISTA with Momentum: Improved Linear Convergence Rate

As is well known in the optimization community, adding a momentum term can accelerate many iterative algorithms. Two classic momentum algorithms were respectively proposed by Polyak [27] and Nesterov [24]. Nesterov's algorithm alternates between the gradient update and the extrapolation, while Polyak's algorithm can be written as an iterative algorithm within the space of gradient updates by simply adding a momentum term. To avoid extra training overhead, we use Polyak's heavy ball scheme and add a momentum term to the ALISTA formula for $k \geq 1$[2] (illustrated in Figure 1):

$$\boldsymbol{x}^{(k+1)} = \eta_{\theta^{(k)}}^{p^{(k)}}\left(\boldsymbol{x}^{(k)} + \gamma^{(k)}\boldsymbol{W}^T(\boldsymbol{b} - \boldsymbol{A}\boldsymbol{x}^{(k)}) + \beta^{(k)}(\boldsymbol{x}^{(k)} - \boldsymbol{x}^{(k-1)})\right) \tag{12}$$

Note that there are other LISTA variants that incorporated additional momentum forms [23, 32], but they did not contribute to provably accelerated convergence rate. Defining constant $\mu$ as the mutual coherence of $\boldsymbol{D}$, i.e., $\mu := \max_{i \neq j} |(\boldsymbol{D}_{:,i})^T\boldsymbol{D}_{:,j}|$, we have the following theorem, which demonstrates that, in the context of (A)LISTA, model (12) can provide a faster linear convergence rate, than that of ALISTA in the same settings.

**Theorem 1** (Convergence of ALISTA-Momentum). *Let $\boldsymbol{x}^{(0)} = \boldsymbol{0}$ and $\{\boldsymbol{x}^{(k)}\}_{k=1}^{\infty}$ be generated by (12). If Assumption 1 holds and $s < (1 + 1/\mu)/2$ and $\sigma$ is small enough, then there exists a uniform sequence of parameter $\{\theta^{(k)}, p^{(k)}, \gamma^{(k)}, \beta^{(k)}\}_{k=1}^{\infty}$ for all $(\boldsymbol{x}^*, \varepsilon) \in \mathcal{X}(B, \underline{B}, s, \sigma)$ such that*

$$\|\boldsymbol{x}^{(k)} - \boldsymbol{x}^*\|_2 \leq C_0 \prod_{t=0}^{k} c^{(t)} + \frac{2sC_W}{1 - 2\mu s + \mu}\sigma, \quad \forall k = 1, 2, \cdots, \tag{13}$$

*where $C_0 > 0$ is a constant depending on $s, B, \mu$ and $C_W = \max_{1 \leq i,j \leq n} |\boldsymbol{W}_{i,j}|$. The convergence rate $c^{(k)} > 0$ satisfies:*

$$c^{(k)} \leq 2\mu s - \mu < 1, \quad \forall k \tag{14}$$

$$c^{(k)} \leq 1 - \sqrt{1 - 2\mu s + \mu} < 2\mu s - \mu, \quad \forall k \geq \frac{\log(\underline{B}) - \log(2C_0)}{\log(2\mu s - \mu)} \tag{15}$$

*and the parameter sequence satisfies*

$$\theta^{(k)} = \mu \sup_{(\boldsymbol{x}^*, \varepsilon) \in \mathcal{X}(B, \underline{B}, s, \sigma)} \left\{\|\boldsymbol{x}^{(k)} - \boldsymbol{x}^*\|_1 + C_W\sigma\right\}, \quad \gamma^{(k)} = 1, \quad \forall k$$

$$\beta^{(k)} \to \left(1 - \sqrt{1 - 2\mu s + \mu}\right)^2, \quad p^{(k)} \to n, \quad \text{as } k \to \infty \tag{16}$$

---

[2]As $k = 0$, we keep the ALISTA formula since $\boldsymbol{x}^{(-1)}$ is not defined.

Despite remaining linearly convergent, our model has an initial rate of $2\mu s - \mu$ that is no worse than the rates in state-of-the-arts [7, 18, 1] and an eventual rate $1 - \sqrt{1 - 2\mu s + \mu}$ that is strictly better than the rate in [7] because $2\mu s - \mu > 1 - \sqrt{1 - 2\mu s + \mu}$ as $2\mu s - \mu < 1$. For example, if $2\mu s - \mu = 0.9$, then our new rate is $1 - \sqrt{1 - 2\mu s + \mu} \approx 0.684$. Moreover, the eventual error $2sC_W\sigma/(1 - 2\mu s + \mu)$ is no worse than that in [7]. Proof details can be found in the supplement.

### 3.3 Instance-Adaptive Parameters: From Linear to Superlinear Convergence

State-of-the-art theoretical results show that LISTA converges with a linear rate [7, 18, 1, 31, 35]. In ALISTA [18], the authors show that linear convergence is a lower bound. This means that one cannot expect a superlinear rate if a uniform set of optimal parameters are learned for a dataset.

One promise to enhance the rate in Theorem 1 is to further introduce instance adaptivity. We establish that, if $\{\theta^{(k)}, p^{(k)}, \gamma^{(k)}, \beta^{(k)}\}_{k=1}^{\infty}$ can be searched for an instance, such instance-optimal parameters lead to superlinear convergence.

**Theorem 2.** *Let $\boldsymbol{x}^{(0)} = \boldsymbol{0}$ and $\{\boldsymbol{x}^{(k)}\}_{k=1}^{\infty}$ be generated by (12). If Assumption 1 holds and $s < (1 + 1/\mu)/2$ and $\sigma$ is small enough, then there exists a sequence of parameters $\{\theta^{(k)}, p^{(k)}, \gamma^{(k)}, \beta^{(k)}\}_{k=1}^{\infty}$ for each instance $(\boldsymbol{x}^*, \varepsilon) \in \mathcal{X}(B, \underline{B}, s, \sigma)$ so that we have:*

$$\|\boldsymbol{x}^{(k)} - \boldsymbol{x}^*\|_2 \leq \sqrt{s}B \prod_{t=0}^{k} \bar{c}^{(t)} + \frac{2sC_W}{1 - 2\mu s + \mu}\sigma, \quad \forall k = 1, 2, \cdots, \tag{17}$$

*where $C_W$ is defined in Theorem 1 and the convergence rate $\bar{c}^{(k)}$ satisfies*

$$\bar{c}^{(k)} \to 0 \quad \text{as} \quad k \to \infty. \tag{18}$$

*To achieve this super-linear convergence rate, parameters are chosen as follows:*

$$\theta^{(k)} = \gamma^{(k)}\Big(\mu\|\boldsymbol{x}^{(k)} - \boldsymbol{x}^*\|_1 + C_W\sigma\Big), \quad \forall k \tag{19}$$

*and $p^{(k)}$ follows (16) for all $k$. The other two parameters $\gamma^{(k)}, \beta^{(k)}$ follow (16) at the initial stage (with small $k$) and they are instance-adaptive as $k$ large enough. Consequently, the model (12) reduces to the conjugate gradient (CG) iteration on the following linear system as $k$ large enough:*

$$\boldsymbol{W}_S^T\big(\boldsymbol{A}_S\boldsymbol{x}_S - \boldsymbol{b}\big) = \boldsymbol{0}, \tag{20}$$

*where $S$ denotes the support of $\boldsymbol{x}^*$ and $\boldsymbol{A}_S, \boldsymbol{W}_S$ denotes, respectively, the sub-matrices of $\boldsymbol{A}, \boldsymbol{W}$ with column mask $S$.*

### 3.4 HyperLISTA: Reducing ALISTA to Tuning Three Hyperparameters

**Adaptive Parameters** Based on Theorem 2, we design the following instance-adaptive parameter formula as the recovery signal $\boldsymbol{x}^{(k)}$ is not accurate enough (equivalently, as $k$ is small):

$$\gamma^{(k)} = 1, \tag{21}$$

$$\theta^{(k)} = c_1\mu\gamma^{(k)}\big\|\boldsymbol{A}^+(\boldsymbol{A}\boldsymbol{x}^{(k)} - \boldsymbol{b})\big\|_1, \tag{22}$$

$$\beta^{(k)} = c_2\mu\,\|\boldsymbol{x}^{(k)}\|_0, \tag{23}$$

$$p^{(k)} = c_3\min\left(\log\left(\frac{\|\boldsymbol{A}^+\boldsymbol{b}\|_1}{\|\boldsymbol{A}^+(\boldsymbol{A}\boldsymbol{x}^{(k)} - \boldsymbol{b})\|_1}\right), n\right), \tag{24}$$

where $c_1 > 0, c_2 > 0, c_3 > 0$ are hyper-parameters to tune.

The step size formula (21) stems from the conclusion (16) as Theorem 2 suggests. The threshold $\theta^{(k)}$ in (19) cannot be directly used due to its dependency on the ground truth $\boldsymbol{x}^*$. In (22), we use $\|\boldsymbol{A}^+(\boldsymbol{A}\boldsymbol{x}^{(k)} - \boldsymbol{b})\|_1$ to estimate (19) because

$$\big\|\boldsymbol{A}^+(\boldsymbol{A}\boldsymbol{x}^{(k)} - \boldsymbol{b})\big\|_1 = \big\|\boldsymbol{A}^+\boldsymbol{A}(\boldsymbol{x}^{(k)} - \boldsymbol{x}^*) - \boldsymbol{A}^+\varepsilon\big\|_1 \approx \mathcal{O}\big(\|\boldsymbol{x}^{(k)} - \boldsymbol{x}^*\|_1\big) + \mathcal{O}\big(\sigma\big).$$

Similar thresholding schemes are studied in [37].

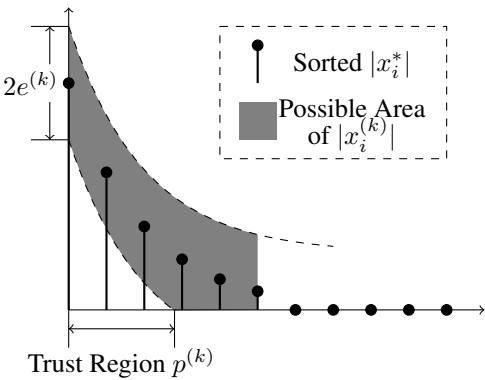

Figure 2: Choice of support selection

The momentum rule (23) is obtained via equation (16). The momentum factor $\beta^{(k)}$ is asymptotically equal to $\left(1 - \sqrt{1 - 2\mu s + \mu}\right)^2$, which is a monotonic increasing function w.r.t. $s$, the number of nonzeros in the ground truth sparse vector $\boldsymbol{x}^*$. Therefore, $\ell_0$ norm of $\boldsymbol{x}^{(k)}$ is a natural choice to approximate $\beta^{(k)}$, considering that $\boldsymbol{x}^{(k)}$ will gradually converge to $\boldsymbol{x}^*$ and hence $\|\boldsymbol{x}^{(k)}\|_0$ to $\|\boldsymbol{x}^*\|_0$.

The support selection rule (24) can be explained as follows. First we sort the magnitude of $|x_i^*|$ as in Figure 2. At the $k$-th layer, given $e^{(k)} = \|\boldsymbol{x}^{(k)} - \boldsymbol{x}^*\|_\infty$, each element in $\boldsymbol{x}^{(k)}$ can be bounded as

$$|x_i^*| - e^{(k)} \le |x_i^{(k)}| \le |x_i^*| + e^{(k)}, \quad \forall i.$$

Such area is shown as the gray area in Figure 2. As we discussed in Section 2 following definition (6), the support selection is a scheme to select those elements that can be trusted as true support. The "trust region" in Figure 2 illustrates such entries. Intuitively, the trust region should be larger as the error $e^{(k)}$ gets smaller. We propose to use the logarithmic function of the error to estimate $p^{(k)}$.

$$p^{(k)} \approx c_3 \log\left(\frac{\|\boldsymbol{x}^*\|_\infty}{e^{(k)}}\right) \approx c_3 \log\left(\frac{\|\boldsymbol{A}^+ \boldsymbol{b}\|_1}{\|\boldsymbol{A}^+(\boldsymbol{A}\boldsymbol{x}^{(k)} - \boldsymbol{b})\|_1}\right) \tag{25}$$

The hyperparameter $0 < c_3 < 1$ is to tune based on the distribution of $\boldsymbol{x}^*$. With an upperbound $n$ (the size of the vector $\boldsymbol{x}^*$), we obtain (24). Actually (25) can be mathematically derived by assuming the magnitude of the nonzero entries of $\boldsymbol{x}^*$ are normally distributed. (The derivation can be found in the supplement.) As $k \to \infty$, it holds that $e^{(k)} \to 0$ and $p^{(k)} \to n$, which means that the operator $\eta_{\theta^{(k)}}^{p^{(k)}}$ is getting close to a hard-thresholding operator. The support selection scheme proposed in LISTA-CPSS [7] also follows such principle but it requires more prior knowledge. Compared to that, formula (24) is much more automatic and self-adaptive.

**Switching to Conjugate Gradients** Theorem 2 suggests that one may call conjugate gradient (CG) to obtain faster convergence as the recovery signal $\boldsymbol{x}^{(k)}$ is accurate enough. In practice, we choose to switch to CG as $k$ is large enough such that $p^{(k)}$ is large enough. The linear system (20) to be solved by CG depends on the support of $\boldsymbol{x}^*$ and we estimate the support by the support of $\boldsymbol{x}^{(k)}$ since we assume $\boldsymbol{x}^{(k)}$ is accurate enough when we call CG. Finally, HyperLISTA is described in Algorithm 1.

### 3.5 Hyperparameter-Tuning Options for HyperLISTA

HyperLISTA has reduced the complexity of the learnable parameters to nearly the extreme: only three. While backpropagation remains to be a viable option to solve them, it becomes an over-kill to solve just three variables by passing gradients through tens of neural network layers. Besides, the hyperparameter $c_3$ in (23) decides the ratio of elements that will be selected into the support and hence passes the thresholding function; that makes $c_3$ non-differentiable in the computation graph.

Instead, we are allowed to seek simpler, even gradient-free methods to search for the parameters. We also empirically find HyperLISTA has certain robustness to perturbing the found values of $c_1, c_2, c_3$, which also encourages us to go with less precise yet much cheaper search methods.

**Algorithm 1:** HyperLISTA with tuned $c_1, c_2, c_3$

---

**Input:** Observation $\boldsymbol{b}$, dictionary $\boldsymbol{A}$, hyperparameters $c_1, c_2, c_3$.
**Initialize :** Let $\boldsymbol{x}^{(0)} = \boldsymbol{0}$.
1 Calculate $\boldsymbol{D}$ and $\boldsymbol{G}$ with (10), set $\boldsymbol{W} = (\boldsymbol{G}^T \boldsymbol{G}) \boldsymbol{A}$.
2 Calculate $\mu$ with $\mu = \max_{i \neq j} |(\boldsymbol{D}^T \boldsymbol{D})_{i,j}|$.
3 **for** $j = 0, 1, 2, \ldots$ *until convergence* **do**
4 $\quad$ Conduct iteration (12) with parameters defined in (22 - 24).
5 $\quad$ **if** $p^{(k)}$ *is large enough* **then**
6 $\quad\quad$ **break**
7 $\quad$ **end**
8 **end**
9 Set $S = \mathrm{supp}(\boldsymbol{x}^{(k)})$, call the conjugate gradient algorithm to solve linear system (20).
**Output:** $\hat{\boldsymbol{x}}$, the result of the conjugate gradient.

---

In this work, we try replacing the backpropagation-based training with the vanilla grid search for learning HyperLISTA. Besides the ultra-light parameterization, another enabling factor is the low evaluation cost on fitness of hyperparameters: just running inference on one minibatch of training samples with them, and no gradient-based training needed. From another aspect, HyperLISTA can be viewed as an iterative algorithm instead of an unfolded neural network. It can be optimized directly on the unseen test data, as long as one wants to afford the cost of re-searching hyperparameters on each dataset, whose cost is still much lower than training using back-propagation.

Specifically, we we first use a coarse grid to find an "interested region" of hyperparameters, and then zoom-in with a fine-grained grid. Details are to be found in Section 4.2, and we plan to try other options such as bayesian optimization [12] in future work.

## 4 Numerical experiments

**Experiment settings** We use synthesized datasets of 51,200 samples for training, 2,048 validation and 2,048 testing. We follow previous works [7, 18] to use a problem size of $(m, n) = (250, 500)$. The elements in the dictionary $A$ are sampled from i.i.d. standard Gaussian distribution and we then normalize $A$ to have unit $\ell_2$ column norms. The sparse vectors $x^*$ are sampled from $N(0, \sigma^2) \cdot Bern(p)$, where $N(\mu, \sigma^2)$ represents the normal distribution and $Bern(p)$ the Bernoulli distribution with probability $p$ to take value 1. By default, we choose $\sigma = 1$ and $p = 0.1$, meaning that around 10% of the elements are non-zero whose magnitudes further follows a standard Gaussian distribution. The additive noise $\varepsilon$ in (1) follows Gaussian distribution $N(0, \sigma_\varepsilon^2)$. The noise level is measured by signal-to-noise ratio (SNR) in decibel unit. Note that we can change the values of $\sigma$, $p$ and $\sigma_\varepsilon$ during testing to evaluate the adaptivity of models. In all experiments, the model performance is measured by normalized mean squared error (NMSE, same as defined in [7, 18]) in decibel unit. All models are unrolled and truncated to 16 layers for training and testing following the typical setting in [7, 18, 31], unless otherwise specified. Besides synthesized data, we also evaluate HyperLISTA on a compressive sensing task using natural images. The results are presented in Appendix B due to limited space.

**Comparison Methods:** In this section, we compare with the original *LISTA* [15] formulated in (2), *ALISTA* formulated in (4) and its variant with momentum acceleration denoted as *ALISTA-MM* (12). A suffix *-Symm* will be added if a model adopts the symmetric parameterization introduced in Section 3.1. We use *HyperLISTA(-Full)* to represent our proposed method in which we optimize all three hyperparameters $c_1, c_2, c_3$ in (22) - (24) using grid search. In contrast, we can use back-propagation to train $c_1$ and $c_2$ and leave $p^{(k)}$ manually selected as in ALISTA because the loss function is not differentiable with respect to $p^{(k)}$. The resulting model is denoted as *HyperLISTA-BP*. Other baselines include *Ada-LISTA* [1] and *NA-ALISTA* [4].

### 4.1 Validation of symmetric matrix parameterization and Theorem 1

We first validate the efficacy of the symmetric matrix parameterization and Theorem 1, showing the acceleration brought by the momentum term. For this validation, we compare four models: ALISTA,

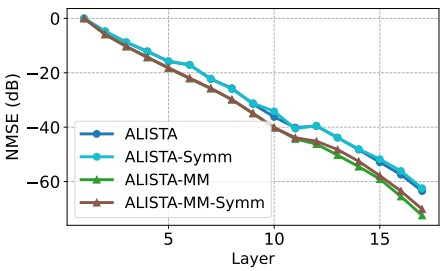

Figure 3: Effect of symmetric matrix parameterization and momentum. Momentum provides acceleration for either parameterization.

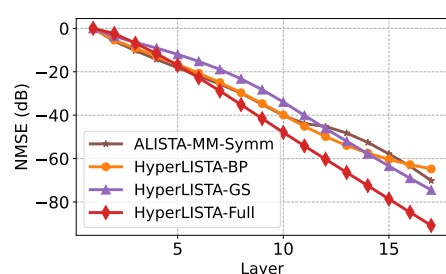

Figure 4: Back-propagation and grid search based training, compared with HyperLISTA-Full with $c_1, c_2, c_3$ searched.

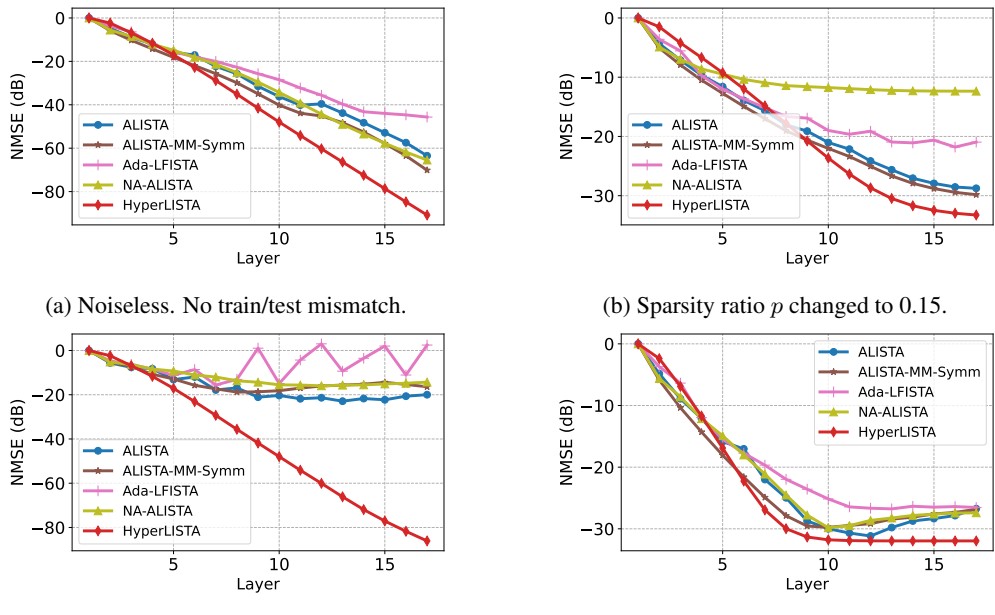

(a) Noiseless. No train/test mismatch.

(b) Sparsity ratio $p$ changed to 0.15.

(c) Variance $\sigma$ of non-zero elements changed to 2.

(d) Noise level changed to SNR=30dB.

Figure 5: Adaptivity experiments. All models are trained in the noiseless case with $p = 0.1$, $\sigma = 1.0$, shown in subfigure (a). We directly apply the models trained in (a) to three different settings, which changes $p$ to 0.15, $\sigma$ to 2 and SNR of the noises to 30 respectively, shown in subfigures (b), (c), (d).

ALISTA-Symm, ALISTA-MM and ALISTA-MM-Symm. We train them on the same dataset and compare their performance measured by NMSE on the testing set. Here, the support selection ratios are selected by following the recommendation in the official implementation of [18]. Results are shown in Figure 3[3]. As we can see from Figure 3, the symmetric matrix parameterization results in almost no performance drop, corroborating the validity of the new parameterization. On the other hand, introducing momentum to both parameterization brings significant improvement in NMSE.

## 4.2 Validation of instance-optimal parameter selection and grid search

In this subsection, we show that the adaptive parameterization in Eqns (22) - (24), i.e., HyperLISTA, can perform as well as or even better than the uniform parameterization in (12), and we also show the efficacy of the grid search method. We optimize HyperLISTA-Full using grid search and train HyperLISTA-BP with backpropagation-based method (SGD), respectively. For a fair comparison with HyperLISTA-BP, we introduce a variant of HyperLISTA-Full in which we only grid-search $c_1$

---

[3]Here ALISTA achieves a worse NMSE than that in [18]. We attribute this to the finite-size training set that we use here instead of unlimited training set in [18].

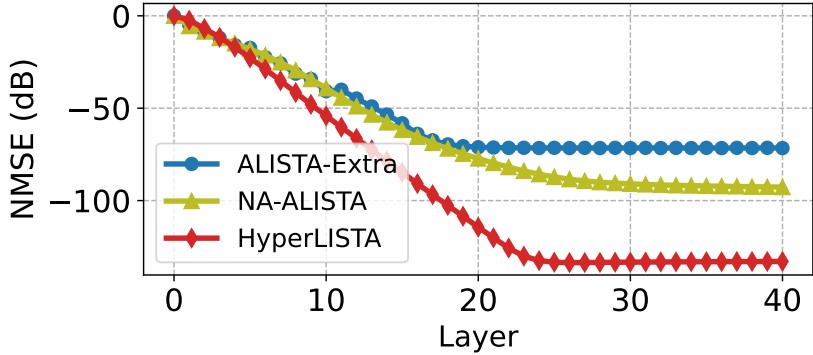

Figure 6: Direct unrolling to 40 layers (all models are trained with 16 layers).

and $c_2$ in (22) and (23) and use the same $p^{(k)}$ as manually selected in HyperLISTA-BP. We denote this variant as *HyperLISTA-GS*. Figure 4 show that grid search can find even better solutions than backpropagation and the uniform parameterization. When we further use grid search for searching $c_1, c_2, c_3$ simultaneously (HyperLISTA-Full), there is more performance boost.

### 4.3 Comparison of adaptivity with previous work

We conduct experiments when there exists mismatch between the training and testing. We instantiate training/testing mismatch in terms of different sparsity levels in the sparse vectors $x^*$, different distributions of non-zero elements in $x^*$, and different additive noise levels. We compare HyperLISTA with ALISTA [18] and its variant ALISTA-MM-Symm, Ada-LFISTA [1] and NA-ALISTA [4] which also dynamically generates parameters per input using an external LSTM. Figure 5 show that in the original and all three mismatched cases, HyperLISTA achieves the best performance, especially showing excellent robustness to the non-zero magnitude distribution change (Fig. 5c). Note our computational overhead is also much lower than using LSTM.

**Directly unrolling HyperLISTA to more layers** Another hidden gem in HyperLISTA is that it learns iteration-independent hyperparameters, which, once searched and trained (on some fixed layer number), can be naturally extended to an arbitrary number of layers. Although NA-ALISTA [4] can also be applied to any number of layers, it has worse performance and adaptivity compared to HyperLISTA, as we compare in Figure 6.

We also compare HyperLISTA with another baseline of 40-layer ALISTA (named as *ALISTA-Extra*), which is directly extrapolated from a 16-layer ALISTA model by reusing the last layer parameters. We can clearly observe that ALISTA cannot directly scale with more layers unrolled.

## 5 Conclusions and Discussions of Broad Impact

In this paper, we propose a ultra-light weight unrolling network, *HyperLISTA*, for solving sparse linear inverse problems. We first introduce a new ALISTA parameterization and augment it with momentum, and then propose an adaptive parameterization which reduces the training of HyperLISTA to only tuning three **instance- and layer-invariant** hyperparameters. HyperLISTA is theoretically proved and also empirically observed to have super-linear convergence rate as it uses instance-optimal parameters. Compared to previous work, HyperLISTA achieves faster convergence on the seen distribution (i.e., the training data distribution) and shows better adaptivity on unseen distributions.

As a limitation, we do not consider perturbations in dictionaries in this work yet. We also find that HyperLISTA, as a compact model similar to ALISTA [18], may fail to perform well on signals with complicated structures such as real images, where the underlying sparse linear model (1) becomes oversimplified and no longer holds well. We will investigate these further in the future.

We do not see that this work will impose any social risks to the society. Besides the intellectual merits, the largest potential societal impact of this work is that unrolling models can be trained more efficiently and meanwhile more adaptive, increasing the reusability.

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
