# A  Empirical Observations on Superlinear Convergence

In this section, we empirically verify the superlinear convergence of HyperLISTA with the switch to conjugate gradient iterations. We apply HyperLISTA to problems with a smaller size: $m = 50, n = 100$. The non-zero elements in the sparse vectors take value 1, instead of following Gaussian distribution in previous experiments. Specifically, we discard support selection in this experiment. Note that setting $c_3 = 0$ in Equation (6), which corresponds to HyperLISTA without support selection, is also in the search space and hence a reasonable option. In this case, we switch to conjugate gradient iterations when the recovered sparse vectors converge to fixed supports for 10 iterations and stop the algorithm as the error is smaller than $-250$ dB. The supports are estimated by filtering the non-zero elements in the recovered vectors that are smaller than 10% of the maximal magnitudes.

In a single sample testing case shown in Figure 7, the algorithm switches to conjugate gradient iterations after the 12th layer and hence we see an accelerated convergence afterwards. This superlinear convergence rate is also observable on a set of 100 testing samples as shown in Figure 8.

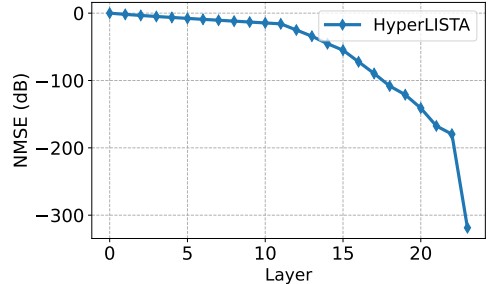
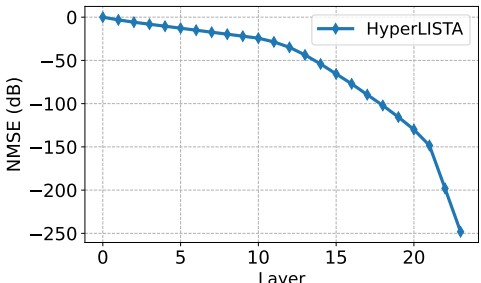

Figure 7: Convergence of HyperLISTA on a single sample.

Figure 8: Convergence of HyperLISTA on 100 testing samples.

# B  Compressive Sensing Experiments on Natural Images

In this section, we conduct a set of compressive sensing experiments on natural images, following a similar setting in [7]. We randomly extract 16x16 patches (flatten into 256-dim vectors then) from the BSD500 dataset [20] to form a training set of 51,200 patches, a validation and a test set of 2,048 patches respectively.

We compressively measure the image signals (256-dim vectors) using a random Gaussian matrix $\mathbf{\Phi} \in \mathbb{R}^{128 \times 256}$. We do not add measuring noises. We apply a dictionary learning algorithm in [34] to the training patches of BSD to learn a matrix $\mathbf{T} \in \mathbb{R}^{256 \times 512}$, where we assume any patch in a natural image can be represented by a sparse combination of the columns in $\mathbf{T}$. We then use $\mathbf{A} = \mathbf{\Phi T}$ as the dictionary $\mathbf{A}$ in (1). In summary, we denote the image patch as $\mathbf{f}$. The measurement we observe, which is also the input to the network, is generated by $\mathbf{b} = \mathbf{\Phi f}$. We assume that there exists a sparse vector $\mathbf{x}^*$ that satisfies $\mathbf{f} \approx \mathbf{T x}^*$ and hence $\mathbf{b} \approx \mathbf{A x}^* = \mathbf{\Phi T x}^*$. We apply different methods to recover $\mathbf{x}^*$ from measurement $b$ and then reconstruct the image patch by multiplying the recovered sparse vector with $\mathbf{T}$.

Because we do not have access to the underlying ground truth $\mathbf{x}^*$ in this real-world scenario, we instead use the LASSO objective function as the loss function for training (of LISTA/ALISTA) or searching (of HyperLISTA), i.e.,

$$Loss = \mathbb{E}\left[\frac{1}{2}\|\mathbf{b} - \mathbf{A}\hat{\mathbf{x}}(\mathbf{b})\|_2^2 + \lambda\|\hat{\mathbf{x}}(\mathbf{b})\|_1\right], \tag{26}$$

where $\hat{\mathbf{x}}(b)$ is the sparse vector recovered from the measurement $\mathbf{b}$ and $\lambda$ is the sparsity regularization coefficient. We set $\lambda = 0.05$ in the experiment. We then reconstruct the image signal by $\hat{\mathbf{f}} = \mathbf{T}\hat{\mathbf{x}}$.

We train LISTA and ALISTA [18], and perform grid search to find optimal hyperparameters in HyperLISTA. These three models all have 16 layers. The model performance is evaluated by the average PSNR of the reconstructed images on the standard testing images in Set 11 used in Recon-Net

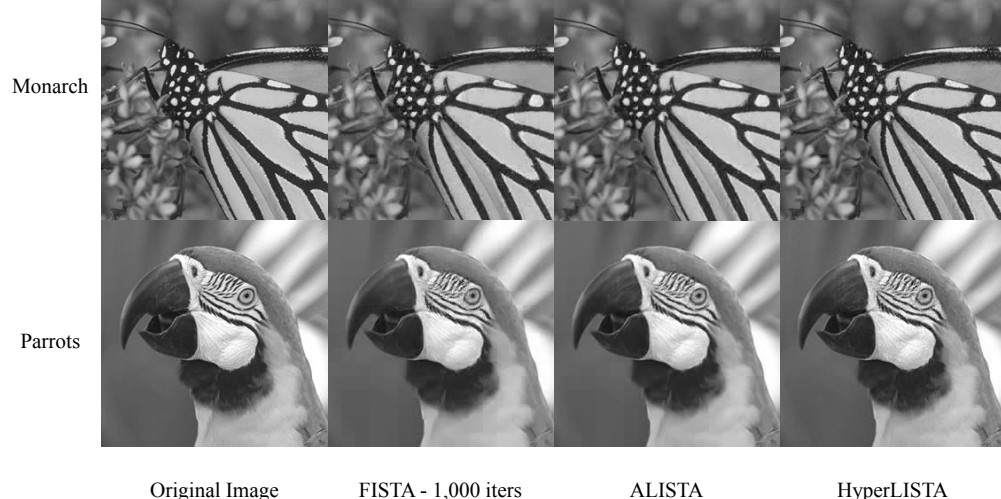

|  | Original Image | FISTA - 1,000 iters | ALISTA | HyperLISTA |

Figure 9: We visually present the recovered images using 1,000 iterations of FISTA and 16-layer ALISTA and HyperLISTA. We can see that HyperLISTA has much less artifacts at the boundaries of patches.

| Method | FISTA 16 iters | FISTA 1,000 iters | LISTA | Recon-Net | ALISTA | HyperLISTA |
|---|---|---|---|---|---|---|
| PSNR (dB) | 29.87 | 30.92 | 30.85 | 31.39 | 32.24 | 33.46 |

[17]. The results are shown in the table below, where we also compare with FISTA and Recon-Net. HyperLISTA outperforms other baselines by clear margins. Note that the performance of LISTA is lower than that reported in [7] where 4,000,000 patches are used for training, while we only use 51,200 training patches. We provide visual results of the recovered images in Figure 9, where we can also see HyperLISTA has less patching artifacts compared to ALISTA and FISTA.

## C    Numerical Algorithm to Calculate Symmetric Dictionary

In this section, we describe how we solve the optimization problem (10):

$$\min_{\boldsymbol{G},\boldsymbol{D}} \|\boldsymbol{D}^T\boldsymbol{D} - \boldsymbol{I}\|_F^2 + \frac{1}{\alpha}\|\boldsymbol{D} - \boldsymbol{G}\boldsymbol{A}\|_F^2, \quad \text{s.t. } \text{diag}(\boldsymbol{D}^T\boldsymbol{D}) = \boldsymbol{1}.$$

Inspired by [19], we use an alternative update algorithm to solve the above problem. Specifically, we first fix $\boldsymbol{G}$ and update $\boldsymbol{D}$ with projected gradient descent (PGD):

$$\boldsymbol{D} \leftarrow \mathcal{P}\Big(\boldsymbol{D} - \zeta\boldsymbol{D}\big(\boldsymbol{D}^T\boldsymbol{D} - \boldsymbol{I}\big) - \frac{\zeta}{\alpha}\big(\boldsymbol{D} - \boldsymbol{G}\boldsymbol{A}\big)\Big), \tag{27}$$

where $\mathcal{P}$ is the projection operator on the constraint $\text{diag}(\boldsymbol{D}^T\boldsymbol{D}) = \boldsymbol{1}$: normalizing each column of $\boldsymbol{D}$. Scalar $\zeta > 0$ is the step size. Secondly, we fix $\boldsymbol{D}$ and calculate the minimizer of $\boldsymbol{G}$ with closed-form solution:

$$\boldsymbol{G} \leftarrow \boldsymbol{D}\boldsymbol{A}^+, \tag{28}$$

Directly using such alternative update algorithm is usually slow when $\alpha$ is small since the condition number is large. However, if we choose $\alpha$ as a large number, the difference between $\boldsymbol{D}$ and $\boldsymbol{G}\boldsymbol{A}$ is not enough penalized. Thus, we use first set $\alpha$ as a large number and solve (10) with (27) and (28). As long as we detect convergence, we tune $\alpha$ as a smaller number. Repeat the procedure until $\boldsymbol{D} \approx \boldsymbol{G}\boldsymbol{A}$. The whole algorithm is described in Algorithm 2.

**Algorithm 2:** Dictionary Solver

**Input:** The original dictionary $\boldsymbol{A}$.
**Initialize:** $\boldsymbol{D} = \boldsymbol{A}, \boldsymbol{G} = \boldsymbol{I}$. $\zeta = \alpha = 0.1$.
1 **for** $j = 0, 1, 2, \ldots$ *until convergence* **do**
2     Update $\boldsymbol{D}$ with (27).
3     Update $\boldsymbol{G}$ with (28).
4     Calculate $f_1 = \|\boldsymbol{D}^T\boldsymbol{D} - \boldsymbol{I}\|_F^2$.
5     Calculate $f_2 = \|(\boldsymbol{G}\boldsymbol{A})^T\boldsymbol{G}\boldsymbol{A} - \boldsymbol{I}\|_F^2$
6     **if** *Two consecutive $f_1$s are close enough* **then**
7        $\alpha \leftarrow 0.1\alpha$.
8        $\zeta \leftarrow 0.1\zeta$.
9        **if** *$f_1$ and $f_2$ are close enough* **then**
10           **break**
11        **end**
12     **end**
13 **end**
**Output:** $\boldsymbol{D}$

## D    Proof of Theorem 1

*Proof.* At the beginning of the proof, we define some constants depending on $\mu, s, B, \underline{B}$:

$$\bar{\beta} := \left(1 - \sqrt{1 - 2\mu s + \mu}\right)^2, \tag{29}$$

$$C_0 := \max\left(1, \frac{5(1 + \bar{\beta})}{\left(4\bar{\beta} - (\bar{\beta} - \mu s + \mu)^2\right)(2\mu s - \mu)}\right)\sqrt{s}B, \tag{30}$$

$$\widetilde{K}_0 := \left\lceil \frac{\log(\underline{B}) - \log(2C_0)}{\log(2\mu s - \mu)} \right\rceil + 1, \tag{31}$$

In the statement of the theorem, we assume "noise level $\sigma$ small enough." Here we give the specific condition on $\sigma$:

$$\sigma \le \min\left(\frac{B}{2}, \sqrt{s}B(2\mu s - \mu)^{\widetilde{K}_0}\right)\frac{1 - 2\mu s + \mu}{2sC_W}. \tag{32}$$

Then we define the parameters we choose in the proof:

$$\theta^{(k)} = \mu \sup_{(\boldsymbol{x}^*, \varepsilon) \in \mathcal{X}(B, \underline{B}, s, \sigma)} \left\{\|\boldsymbol{x}^{(k)} - \boldsymbol{x}^*\|_1 + C_W\sigma\right\}, \tag{33}$$

$$\gamma^{(k)} = 1, \tag{34}$$

$$\beta^{(k)} = \begin{cases} 0, & k \le \widetilde{K}_0 \\ \bar{\beta}, & k > \widetilde{K}_0 \end{cases} \tag{35}$$

$$p^{(k)} = \min\left(\frac{k}{\widetilde{K}_0}n, n\right), \tag{36}$$

It is easy to check that the conclusion (16) in the theorem is satisfied.

The whole proof scheme consists of several steps: proving no false positives for all $k$; proving the conclusion for $k \le \widetilde{K}_0$; and proving the conclusion for $k > \widetilde{K}_0$.

**Step 1: no false positives.** Firstly, we take an $(\boldsymbol{x}^*, \varepsilon) \in \mathcal{X}(B, \underline{B}, s, \sigma)$. Let $S = \mathrm{support}(\boldsymbol{x}^*)$. We want to prove by induction that, as long as (33) holds, $x_i^{(k)} = 0, \forall i \notin S, \forall k$ (no false positives). The statement in the theorem gives $\boldsymbol{x}^{(0)} = \boldsymbol{0}$. We follow the same proof line with that in [7] and obtain $\boldsymbol{x}^{(1)} = \boldsymbol{0}$. Fixing $k \ge 1$, and assuming $x_i^{(t)} = 0, \forall i \notin S, t = 0, 1, 2, \cdots, k$, we have

$$\begin{aligned} x_i^{(k+1)} &= \eta_{\theta^{(k)}}^{p^{(k)}}\left(x_i^{(k)} - (\boldsymbol{W}_{:,i})^T(\boldsymbol{A}\boldsymbol{x}^{(k)} - \boldsymbol{b}) + \beta^{(k)}\left(x_i^{(k)} - x_i^{(k-1)}\right)\right) \\ &= \eta_{\theta^{(k)}}^{p^{(k)}}\left(-\sum_{j \in S}(\boldsymbol{W}_{:,i})^T\boldsymbol{A}_{:,j}(x_j^{(k)} - x_j^*) + (\boldsymbol{W}_{:,i})^T\varepsilon\right), \quad \forall i \notin S. \end{aligned}$$

By the definition of $\mu$:

$$\mu = \max_{i \neq j} \left| (\boldsymbol{D}_{:,i})^T \boldsymbol{D}_{:,j} \right| = \max_{i \neq j} \left| (\boldsymbol{D}^T \boldsymbol{D})_{i,j} \right| = \max_{i \neq j} \left| (\boldsymbol{W}^T \boldsymbol{A})_{i,j} \right|, \tag{37}$$

we have

$$\mu \|\boldsymbol{x}^{(k)} - \boldsymbol{x}^*\|_1 \geq \left| -\sum_{j \in S} (\boldsymbol{W}_{:,i})^T \boldsymbol{A}_{:,j} (x_j^{(k)} - x_j^*) \right|, \quad \forall i \notin S.$$

By the definition of $C_W$, we have

$$C_W \sigma \geq \max_{1 \leq j \leq m} |W_{j,i}| \cdot \|\varepsilon\|_1 \geq \left| (\boldsymbol{W}_{:,i})^T \varepsilon \right|, \quad \forall i = 1, \cdots, n.$$

Then we can obtain a lower bound for the threshold $\theta^{(k)}$:

$$\theta^{(k)} \geq \mu \|\boldsymbol{x}^{(k)} - \boldsymbol{x}^*\|_1 + C_W \sigma \geq \left| -\sum_{j \in S} (\boldsymbol{W}_{:,i})^T \boldsymbol{A}_{:,j} (x_j^{(k)} - x_j^*) + (\boldsymbol{W}_{:,i})^T \varepsilon \right|,$$

which implies $x_i^{(k+1)} = 0, \forall i \notin S$ by the definition of $\eta_{\theta^{(k)}}^{p^{(k)}}$ in (6). By induction, we have

$$x_i^{(k)} = 0, \forall i \notin S, \quad \forall k. \tag{38}$$

**Step 2: analysis for $k \leq \widetilde{K}_0$.** Since $\beta^{(k)} = 0$ as $k \leq \widetilde{K}_0$, ALISTA-MM reduces to ALISTA. One can easily follow the proof line of Theorem 3 in [7] to obtain

$$\|\boldsymbol{x}^{(k)} - \boldsymbol{x}^*\|_2 \leq sB(2\mu s - \mu)^k + \frac{2sC_W}{1 + \mu - 2\mu s} \sigma$$

Here we introduce a new proof method to obtain a better bound:

$$\|\boldsymbol{x}^{(k)} - \boldsymbol{x}^*\|_2 \leq \sqrt{s}B(2\mu s - \mu)^k + \frac{2sC_W}{1 + \mu - 2\mu s} \sigma. \tag{39}$$

To start our proof, we combine ALISTA-Momentum formula (12) and (38) and obtain

$$\boldsymbol{x}_S^{(k+1)} = \eta_{\theta^{(k)}}^{p^{(k)}} \left( \boldsymbol{x}_S^{(k)} - (\boldsymbol{W}_{:,S})^T (\boldsymbol{A}_{:,S} \boldsymbol{x}_S^{(k)} - \boldsymbol{b}) + \beta^{(k)} (\boldsymbol{x}_S^{(k)} - \boldsymbol{x}_S^{(k-1)}) \right).$$

With $\boldsymbol{b} = \boldsymbol{A}\boldsymbol{x}^* + \varepsilon$, we have

$$\boldsymbol{x}_S^{(k+1)} = \eta_{\theta^{(k)}}^{p^{(k)}} \left( \boldsymbol{x}_S^{(k)} - (\boldsymbol{W}_{:,S})^T \boldsymbol{A}_{:,S} (\boldsymbol{x}_S^{(k)} - \boldsymbol{x}_S^*) + (\boldsymbol{W}_{:,S})^T \varepsilon + \beta^{(k)} (\boldsymbol{x}_S^{(k)} - \boldsymbol{x}_S^{(k-1)}) \right).$$

To analyze the thresholding operator $\eta_{\theta^{(k)}}^{p^{(k)}}$, we define $\boldsymbol{v}^{(k)}$ elementwisely as (The set $S^{p^{(k)}}$ is defined in (7).):

$$v_i^{(k)} \begin{cases} \in [-1, 1] & \text{if } i \in S, x_i^{(k+1)} = 0 \\ = \text{sign}(x_i^{(k+1)}) & \text{if } i \in S, x_i^{(k+1)} \neq 0, i \notin S^{p^{(k)}}(\boldsymbol{x}^{(k+1)}), \\ = 0 & \text{if } i \in S, x_i^{(k+1)} \neq 0, i \in S^{p^{(k)}}(\boldsymbol{x}^{(k+1)}). \end{cases} \tag{40}$$

Given (40), we have

$$\begin{aligned} \boldsymbol{x}_S^{(k+1)} = \boldsymbol{x}_S^{(k)} - (\boldsymbol{W}_{:,S})^T \boldsymbol{A}_{:,S} (\boldsymbol{x}_S^{(k)} - \boldsymbol{x}_S^*) + \beta^{(k)} (\boldsymbol{x}_S^{(k)} - \boldsymbol{x}_S^{(k-1)}) \\ + (\boldsymbol{W}_{:,S})^T \varepsilon - \theta^{(k)} \boldsymbol{v}^{(k)}, \quad \forall k \geq 1. \end{aligned} \tag{41}$$

Subtracting $\boldsymbol{x}^*$ from the both sides of (41), we obtain

$$\begin{aligned} \boldsymbol{x}_S^{(k+1)} - \boldsymbol{x}_S^* = \left( (1 + \beta^{(k)}) \boldsymbol{I}_S - (\boldsymbol{W}_{:,S})^T \boldsymbol{A}_{:,S} \right) (\boldsymbol{x}_S^{(k)} - \boldsymbol{x}_S^*) - \beta^{(k)} (\boldsymbol{x}_S^{(k-1)} - \boldsymbol{x}_S^*) \\ + (\boldsymbol{W}_{:,S})^T \varepsilon - \theta^{(k)} \boldsymbol{v}^{(k)}. \end{aligned} \tag{42}$$

The first term of the above line can be bounded by

$$\left\| (\boldsymbol{W}_{:,S})^T \varepsilon \right\|_2 \leq \left\| (\boldsymbol{W}_{:,S})^T \varepsilon \right\|_1 \leq \sum_{i \in S} \left| (\boldsymbol{W}_{:,i})^T \varepsilon \right| \leq sC_W \sigma.$$

To analyze the norm of vector $\boldsymbol{v}^{(k)}$, we define

$$S^{(k)}(\boldsymbol{x}^*, \varepsilon) = \left\{ i \mid i \in S, x_i^{(k+1)} \neq 0, i \in S^{p^{(k)}}(\boldsymbol{x}^{(k+1)}) \right\},$$

$$\overline{S}^{(k)}(\boldsymbol{x}^*, \varepsilon) = S \backslash S^{(k)}, \tag{43}$$

where $S^{(k)}$ depends on $\boldsymbol{x}^*$ and $\varepsilon$ because $\boldsymbol{x}^{(k+1)}$ depends on $\boldsymbol{x}^*$ and $\varepsilon$. Given this definition, we have

$$\|\boldsymbol{v}^{(k)}\|_2 \leq \sqrt{|\overline{S}^{(k)}(\boldsymbol{x}^*, \varepsilon)|}.$$

As $k \leq \widetilde{K}_0$, $\beta^{(k)} = 0$ (35), equation (42) implies

$$\|\boldsymbol{x}_S^{(k+1)} - \boldsymbol{x}_S^*\|_2 \leq \left\| \boldsymbol{I}_S - (\boldsymbol{W}_{:,S})^T \boldsymbol{A}_{:,S} \right\|_2 \|\boldsymbol{x}_S^{(k)} - \boldsymbol{x}_S^*\|_2 + sC_W\sigma + \theta^{(k)}\sqrt{|\overline{S}^{(k)}(\boldsymbol{x}^*, \varepsilon)|}, \quad k \leq \widetilde{K}_0.$$

The definition of $\mu$ (37) means each off-diagonal element of matrix $\boldsymbol{W}^T\boldsymbol{A}$ is no greater than $\mu$. Thus, the sum of all off-diagonal elements of matrix $(\boldsymbol{W}^T\boldsymbol{A})_{S,S}$ is no greater than $\mu(|S| - 1)$. The diagonal elements of matrix $\boldsymbol{I}_S - (\boldsymbol{W}^T\boldsymbol{A})_{S,S}$ are actually zeros by the definition of $\boldsymbol{W}$ in (10). Then Gershgorin circle theorem [14] implies

$$\|\boldsymbol{I}_S - (\boldsymbol{W}^T\boldsymbol{A})_{S,S}\|_2 \leq \mu(|S| - 1) \leq \mu(s - 1). \tag{44}$$

Since $x_i^{(k)} = 0, \forall i \notin S$, we have $\|\boldsymbol{x}^{(k)} - \boldsymbol{x}^*\|_2 = \|\boldsymbol{x}_S^{(k)} - \boldsymbol{x}_S^*\|_2$ for all $k$. Taking supremum on both sides of inequality (41) over $(\boldsymbol{x}^*, \varepsilon) \in \mathcal{X}(B, \underline{B}, s, \sigma)$, we have

$$\sup_{(\boldsymbol{x}^*, \varepsilon) \in \mathcal{X}(B, \underline{B}, s, \sigma)} \|\boldsymbol{x}^{(k+1)} - \boldsymbol{x}^*\|_2 \leq (\mu(s - 1)) \sup_{(\boldsymbol{x}^*, \varepsilon) \in \mathcal{X}(B, \underline{B}, s, \sigma)} \|\boldsymbol{x}^{(k)} - \boldsymbol{x}^*\|_2 + sC_W\sigma$$

$$+ \theta^{(k)}\sqrt{\sup_{(\boldsymbol{x}^*, \varepsilon) \in \mathcal{X}(B, \underline{B}, s, \sigma)} |\overline{S}^{(k)}(\boldsymbol{x}^*, \varepsilon)|}.$$

Since $\overline{S}^{(k)}$ is a subset of $S$, it holds that

$$\left| \overline{S}^{(k)}(\boldsymbol{x}^*, \varepsilon) \right| \leq |S| \leq s.$$

By the definition of $\theta^{(k)}$ in (22), it holds that

$$\theta^{(k)} = \sup_{(\boldsymbol{x}^*, \varepsilon) \in \mathcal{X}(B, \underline{B}, s, \sigma)} \left\{ \mu\|\boldsymbol{x}^{(k)} - \boldsymbol{x}^*\|_1 \right\} + C_W\sigma$$

$$\leq \sup_{(\boldsymbol{x}^*, \varepsilon) \in \mathcal{X}(B, \underline{B}, s, \sigma)} \left\{ \mu\sqrt{s}\|\boldsymbol{x}^{(k)} - \boldsymbol{x}^*\|_2 \right\} + C_W\sigma$$

Combining the above inequalities, we get

$$\sup_{\boldsymbol{x}^* \in \mathcal{X}(B, \underline{B}, s)} \|\boldsymbol{x}^{(k+1)} - \boldsymbol{x}^*\|_2 \leq (2\mu s - \mu) \sup_{\boldsymbol{x}^* \in \mathcal{X}(B, \underline{B}, s)} \|\boldsymbol{x}^{(k)} - \boldsymbol{x}^*\|_2 + 2sC_W\sigma.$$

Denoting $e^{(k)} = \sup_{(\boldsymbol{x}^*, \varepsilon) \in \mathcal{X}(B, \underline{B}, s, \sigma)} \|\boldsymbol{x}^{(k)} - \boldsymbol{x}^*\|_2$, applying the above inequality repeatedly as $k = 0, 1, 2, \cdots$, we obtain

$$e^{(k)} \leq (2\mu s - \mu)^k e^{(0)} + 2sC_W\sigma \left( \sum_{t=0}^k (2\mu s - \mu)^t \right)$$

$$\leq (2\mu s - \mu)^k e^{(0)} + \frac{2sC_W}{1 + \mu - 2\mu s}\sigma.$$

Since $\boldsymbol{x}^{(0)} = \boldsymbol{0}$, the term $e^{(0)}$ can be bounded by

$$e^{(0)} = \sup_{(\boldsymbol{x}^*, \varepsilon) \in \mathcal{X}(B, \underline{B}, s, \sigma)} \|\boldsymbol{x}^*\|_2 \leq \sqrt{s}B \leq C_0.$$

Then we conclude with

$$\|\boldsymbol{x}^{(k)} - \boldsymbol{x}^*\|_2 \leq C_0(2\mu s - \mu)^k + \frac{2sC_W}{1 + \mu - 2\mu s}\sigma, \quad \forall k \leq \widetilde{K}_0.$$

**Step 3: analysis for $k > \widetilde{K}_0$.**

Define

$$\delta \boldsymbol{x}_S = \left((\boldsymbol{W}_{:,S})^T \boldsymbol{A}_{:,S}\right)^{-1}(\boldsymbol{W}_{:,S})^T \varepsilon, \quad \overline{\boldsymbol{x}}_S = \boldsymbol{x}_S^* + \delta \boldsymbol{x}_S. \tag{45}$$

It can be proved that

$$\|\delta \boldsymbol{x}_S\|_2 \leq \frac{1}{1 + \mu - \mu s}\|(\boldsymbol{W}_{:,S})^T \varepsilon\|_2 \leq \frac{1}{1 + \mu - \mu s}\|(\boldsymbol{W}_{:,S})^T \varepsilon\|_1 \leq \frac{sC_W}{1 + \mu - \mu s}\sigma. \tag{46}$$

Then (41) can be rewritten as

$$\boldsymbol{x}_S^{(k+1)} = \boldsymbol{x}_S^{(k)} - (\boldsymbol{W}_{:,S})^T \boldsymbol{A}_{:,S}(\boldsymbol{x}_S^{(k)} - \overline{\boldsymbol{x}}_S) + \beta^{(k)}\left(\boldsymbol{x}_S^{(k)} - \boldsymbol{x}_S^{(k-1)}\right) - \theta^{(k)}\boldsymbol{v}^{(k)}.$$

Subtracting $\overline{\boldsymbol{x}}_S$ from both sides of the above formula, we obtain

$$\boldsymbol{x}_S^{(k+1)} - \overline{\boldsymbol{x}}_S = \left((1 + \beta^{(k)})\boldsymbol{I}_S - (\boldsymbol{W}_{:,S})^T \boldsymbol{A}_{:,S}\right)(\boldsymbol{x}_S^{(k)} - \overline{\boldsymbol{x}}_S) - \beta^{(k)}\left(\boldsymbol{x}_S^{(k-1)} - \overline{\boldsymbol{x}}_S\right) - \theta^{(k)}\boldsymbol{v}^{(k)}.$$

Plug $\beta^{(k)} = \bar{\beta}$ for $k > \widetilde{K}_0$ (35) into the above equation. We obtain

$$\underbrace{\begin{bmatrix} \boldsymbol{x}_S^{(k+1)} - \overline{\boldsymbol{x}}_S \\ \boldsymbol{x}_S^{(k)} - \overline{\boldsymbol{x}}_S \end{bmatrix}}_{\boldsymbol{z}^{(k)}} = \underbrace{\begin{bmatrix} (1 + \bar{\beta})\boldsymbol{I}_S - (\boldsymbol{W}_{:,S})^T \boldsymbol{A}_{:,S} & -\bar{\beta}\boldsymbol{I}_S \\ \boldsymbol{I}_S & \boldsymbol{0} \end{bmatrix}}_{\boldsymbol{M}} \underbrace{\begin{bmatrix} \boldsymbol{x}_S^{(k)} - \overline{\boldsymbol{x}}_S \\ \boldsymbol{x}_S^{(k-1)} - \overline{\boldsymbol{x}}_S \end{bmatrix}}_{\boldsymbol{z}^{(k-1)}} - \theta^{(k)}\begin{bmatrix} \boldsymbol{v}^{(k)} \\ \boldsymbol{0} \end{bmatrix}. \tag{47}$$

We claim that $\boldsymbol{M}$ can be decomposed with

$$\boldsymbol{M} = \boldsymbol{T}\Lambda_{\boldsymbol{M}}\boldsymbol{T}^{-1}, \tag{48}$$

where $\boldsymbol{T} \in \mathbb{C}^{2|S| \times 2|S|}$ is nonsingular and $\Lambda_{\boldsymbol{M}} \in \mathbb{C}^{2|S| \times 2|S|}$ is diagonal and they satisfy

$$\|\Lambda_{\boldsymbol{M}}\|_2 = \sqrt{\bar{\beta}},$$

$$\|\boldsymbol{T}\|_2 \leq \sqrt{2 + 2\bar{\beta}}, \tag{49}$$

$$\|\boldsymbol{T}^{-1}\|_2 \leq \frac{\sqrt{2 + 2\bar{\beta}}}{2(4\bar{\beta} - (\bar{\beta} - \mu s + \mu)^2)}.$$

We will prove claims (48) and (49) later in step 4. Note that matrix $\boldsymbol{M}$ and its decomposition (48) are not uniform for different instances that satisfy Assumption 1, but the upper bounds in (49) are **uniform** for all instances that satisfy $|S| \leq s$ and $s \leq (1 + 1/\mu)/2$. This is why we do not only calculate the spectral norm of $\boldsymbol{M}$ as the standard heavy-ball momentum proof: we want to get an **uniform** bound for all instances but not a specific bound for each instance.

Since $\boldsymbol{M}$ is not symmetric, (47) indicates that $\|\boldsymbol{x}_S^{(k)} - \overline{\boldsymbol{x}}_S\|_2$ is NOT monotonically decreasing due to the momentum term. Thus, $\boldsymbol{v}^{(k)} = \boldsymbol{0}$ cannot imply $\boldsymbol{v}^{(k+1)} = \boldsymbol{0}$. We have to estimate the upperbound of $\|\boldsymbol{x}_S^{(k)} - \overline{\boldsymbol{x}}_S\|_2$ by **induction**:

- First we prove that $\boldsymbol{v}^{(k)} = \boldsymbol{0}$ as $k = \widetilde{K}_0$.

- Then we prove that, if we assume $\boldsymbol{v}^{(t)} = \boldsymbol{0}$ for all t satisfies $\widetilde{K}_0 \leq t \leq k(k > \widetilde{K}_0)$, we have

$$\|\boldsymbol{x}_S^{(k+1)} - \overline{\boldsymbol{x}}_S\|_2 \leq C_0(2\mu s - \mu)^{\widetilde{K}_0}\left(1 - \sqrt{1 - 2\mu s + \mu}\right)^{k+1-\widetilde{K}_0} \text{ and } \boldsymbol{v}^{(k+1)} = \boldsymbol{0}.$$

Now we prove the above two points one-by-one.

First we want to show that $\boldsymbol{v}^{(k)} = \boldsymbol{0}$ as $k = \widetilde{K}_0$. The definition of $\widetilde{K}_0$ (31) implies

$$C_0(2\mu s - \mu)^{\widetilde{K}_0} < \underline{B}/2.$$

The assumption on $\sigma$ (32) implies

$$\frac{2sC_W}{1 - 2\mu s + \mu}\sigma \leq \underline{B}/2.$$

As $k = \widetilde{K}_0$, it holds that, for all $i \in S$

$$|x_i^{(k)} - x_i^*| \le \|\boldsymbol{x}^{(k)} - \boldsymbol{x}^*\|_2 \le C_0(2\mu s - \mu)^k + \frac{2sC_W}{1 - 2\mu s + \mu}\sigma < \underline{B}. \tag{50}$$

Definition of $p^{(k)}$ in (36) implies $p^{(k)} = n$ for $k \ge \widetilde{K}_0$. Based on the definition of $\eta$ operator (6) and (7), we have $S^{p^{(k)}} = \{1, 2, \cdots, n\}$ and the operator $\eta_{\theta^{(k)}}^{p^{(k)}}$ is actually a hard thresholding. The above inequality (50), combined with $|x^*|_i \ge \underline{B} > 0$, implies $x_i^{(k)} \ne 0, i \in S$. Since $p^{(k)} = n$, we have $i \in S^{(k)}(\boldsymbol{x}^*, \varepsilon)$ for all $i \in S$ according to (43). In another word, $\boldsymbol{v}^{(k)} = \boldsymbol{0}$ as $k = \widetilde{K}_0$.

Second, we assume assume $\boldsymbol{v}^{(t)} = \boldsymbol{0}$ for all $\widetilde{K}_0 \le t \le k$ and analyze the variables for $k+1$. Equation (47) and the assumption $\boldsymbol{v}^{(t)} = \boldsymbol{0}$ and the matrix decomposition (48) lead to

$$\mathbf{T}^{-1}\boldsymbol{z}^{(t)} = \Lambda_{\mathbf{M}}\mathbf{T}^{-1}\boldsymbol{z}^{(t-1)}, \quad \widetilde{K}_0 \le t \le k.$$

Then we have

$$\mathbf{T}^{-1}\boldsymbol{z}^{(k)} = \left(\Lambda_{\mathbf{M}}\right)^{k+1-\widetilde{K}_0}\mathbf{T}^{-1}\boldsymbol{z}^{(\widetilde{K}_0-1)}.$$

Thus, it holds that

$$\|\boldsymbol{z}^{(k)}\|_2 \le \|\Lambda_{\mathbf{M}}\|_2^{k+1-\widetilde{K}_0}\|\mathbf{T}\|_2\|\mathbf{T}^{-1}\|_2\|\boldsymbol{z}^{(\widetilde{K}_0-1)}\|_2 \le \frac{(1+\bar{\beta})\|\boldsymbol{z}^{(\widetilde{K}_0-1)}\|_2}{4\bar{\beta} - (\bar{\beta} - \mu s + \mu)^2}\left(\sqrt{\bar{\beta}}\right)^{k+1-\widetilde{K}_0}.$$

By (46), (39) and (32), we have

$$\|\boldsymbol{z}^{(\widetilde{K}_0-1)}\|_2 \le 2\|\boldsymbol{x}^{(\widetilde{K}_0-1)} - \overline{\boldsymbol{x}}\|_2 \le 2\|\boldsymbol{x}^{(\widetilde{K}_0-1)} - \boldsymbol{x}^*\|_2 + 2\|\delta\boldsymbol{x}_S\|_2 \le 5\sqrt{s}B(2\mu s - \mu)^{\widetilde{K}_0-1}.$$

Consequently,

$$\|\boldsymbol{x}_S^{(k+1)} - \overline{\boldsymbol{x}}_S\|_2 \le \|\boldsymbol{z}^{(k)}\|_2 \le C_0(2\mu s - \mu)^{\widetilde{K}_0}\left(1 - \sqrt{1 - 2\mu s + \mu}\right)^{k+1-\widetilde{K}_0}.$$

The definition of $\widetilde{K}_0$ (31) gives

$$C_0(2\mu s - \mu)^k < \underline{B}/2, \quad \forall k \ge \widetilde{K}_0.$$

It's easy to check that the following inequality holds for all $s \le (1 + 1/\mu)/2$:

$$1 - \sqrt{1 - 2\mu s + \mu} < 2\mu s - \mu.$$

We conclude that

$$|x_i^{(k+1)} - \overline{x}_i| \le \|\boldsymbol{x}_S^{(k+1)} - \overline{\boldsymbol{x}}_S\|_2 \le C_0(2\mu s - \mu)^{k+1} < \underline{B}/2, \quad \forall i \in S.$$

With similar arguments as (50), we obtain

$$|x_i^{(k+1)} - x_i^*| \le |x_i^{(k+1)} - \overline{x}_i| + |\delta x_i| < \underline{B}/2 + \underline{B}/2 = \underline{B}, \quad \forall i \in S$$

and then $\boldsymbol{v}^{(k+1)} = \boldsymbol{0}$.

The induction proof is finished and we conclude that, for all $k > \widetilde{K}_0$,

$$\|\boldsymbol{x}^{(k)} - \boldsymbol{x}^*\|_2 \le \|\boldsymbol{x}_S^{(k)} - \overline{\boldsymbol{x}}_S\|_2 + \|\delta\boldsymbol{x}_S\|_2$$
$$\le C_0(2\mu s - \mu)^{\widetilde{K}_0}\left(1 - \sqrt{1 - 2\mu s + \mu}\right)^{k+1-\widetilde{K}_0} + \frac{sC_W}{1 + \mu - \mu s}\sigma.$$

As long as we prove (48) and (49), we can finish the whole proof. Proofs of (48) and (49) are actually an estimate of the spectrum of matrix $\mathbf{M}$ and given in "Step 4."

**Step 4: spectrum analysis for $\mathbf{M}$.** Since $\boldsymbol{W}^T\boldsymbol{A}$ is parameterized in a symmetric way (Section 3.1), its submatrix $(\boldsymbol{W}_{:,S})^T\boldsymbol{A}_{:,S}$ is a real symmetric matrix and, consequently, it is orthogonal diagonalizable in the real space:

$$(\boldsymbol{W}_{:,S})^T\boldsymbol{A}_{:,S} = \boldsymbol{U}\Lambda\boldsymbol{U}^T$$

where $\boldsymbol{U} \in \mathbb{R}^{|S| \times |S|}$ is orthogonal and $\boldsymbol{\Lambda} \in \mathbb{R}^{|S| \times |S|}$ is diagonal. The matrix $\boldsymbol{\Lambda}$ is defined as $\boldsymbol{\Lambda} = \mathrm{diag}(\lambda_1, \lambda_2, \cdots, \lambda_{|S|})$ and its elements are singular values of $(\boldsymbol{W}_{:,S})^T \boldsymbol{A}_{:,S}$. Thus, it holds that

$$\mathbf{M} = \begin{bmatrix} \boldsymbol{U}\big((1+\bar{\beta})\boldsymbol{I}_S - \boldsymbol{\Lambda}\big)\boldsymbol{U}^T & -\boldsymbol{U}\big(\bar{\beta}\boldsymbol{I}_S\big)\boldsymbol{U}^T \\ \boldsymbol{U}(\boldsymbol{I}_S)\boldsymbol{U}^T & \mathbf{0} \end{bmatrix}$$

$$= \begin{bmatrix} \boldsymbol{U} & \mathbf{0} \\ \mathbf{0} & \boldsymbol{U} \end{bmatrix} \begin{bmatrix} (1+\bar{\beta})\boldsymbol{I}_S - \boldsymbol{\Lambda} & -\bar{\beta}\boldsymbol{I}_S \\ \boldsymbol{I}_S & \mathbf{0} \end{bmatrix} \begin{bmatrix} \boldsymbol{U}^T & \mathbf{0} \\ \mathbf{0} & \boldsymbol{U}^T \end{bmatrix}.$$

By properly permuting the index, we obtain

$$\mathbf{M} = \begin{bmatrix} \boldsymbol{U} & \mathbf{0} \\ \mathbf{0} & \boldsymbol{U} \end{bmatrix} \mathbf{P} \begin{bmatrix} \mathbf{B}_1 & & & \\ & \mathbf{B}_2 & & \\ & & \cdots & \\ & & & \mathbf{B}_{|S|} \end{bmatrix} \mathbf{P}^{-1} \begin{bmatrix} \boldsymbol{U}^T & \mathbf{0} \\ \mathbf{0} & \boldsymbol{U}^T \end{bmatrix} \tag{51}$$

where $\mathbf{P}$ is a permutation operator with $\|\mathbf{P}\|_2 = 1$ and the blocks $\mathbf{B}_i$ is defined as

$$\mathbf{B}_i = \begin{bmatrix} 1 + \bar{\beta} - \lambda_i & -\bar{\beta} \\ 1 & 0 \end{bmatrix}, \quad \forall i = 1, 2, \cdots, |S|. \tag{52}$$

The two eigenvalues and eigenvectors of $\mathbf{B}_i$ can be calculated explicitly:

$$\lambda_{\mathbf{B}_i,1} = \frac{1}{2}\Big(1 + \bar{\beta} - \lambda_i + \sqrt{(1+\bar{\beta}-\lambda_i)^2 - 4\bar{\beta}}\Big)$$

$$\lambda_{\mathbf{B}_i,2} = \frac{1}{2}\Big(1 + \bar{\beta} - \lambda_i - \sqrt{(1+\bar{\beta}-\lambda_i)^2 - 4\bar{\beta}}\Big)$$

$$\mathbf{v}_{\mathbf{B}_i,1} = \begin{bmatrix} \lambda_{\mathbf{B}_i,1} \\ 1 \end{bmatrix}, \quad \mathbf{v}_{\mathbf{B}_i,2} = \begin{bmatrix} \lambda_{\mathbf{B}_i,2} \\ 1 \end{bmatrix}$$

Define

$$\delta(\lambda; \bar{\beta}) = (1 + \bar{\beta} - \lambda)^2 - 4\bar{\beta}$$

that is a one dimensional quadratic function for $\lambda$. It is convex and its critical point is $\lambda_\star = 1 + \bar{\beta}$. It has two zero points $(1 \pm \sqrt{\bar{\beta}})^2$. According to (44), the eigenvalues of $(\boldsymbol{W}_{:,S})^T \boldsymbol{A}_{:,S}$ are bounded within the following interval

$$1 - \mu(s-1) \le \lambda_i \le 1 + \mu(s-1), \quad \forall i = 1, 2, \cdots, |S|.$$

It's easy to check that the maximum of $\delta(\lambda; \bar{\beta})$ function is taken at the left boundary:

$$\arg\max_{\lambda \in [1-\mu(s-1), 1+\mu(s-1)]} \delta(\lambda; \bar{\beta}) = 1 - \mu(s-1).$$

Since it holds that

$$1 - \mu(s-1) > 1 - 2\mu s + \mu = (1 - \sqrt{\bar{\beta}})^2,$$

we obtain

$$\delta\Big(1 - \mu(s-1); \bar{\beta}\Big) < \delta\Big((1 - \sqrt{\bar{\beta}})^2; \bar{\beta}\Big) = 0.$$

Thus,

$$\delta(\lambda_i; \bar{\beta}) < 0, \forall i = 1, 2, \cdots, |S|,$$

which implies that eigenvalues of $\mathbf{B}_i$ has nonzero imaginary part and the two eigenvalues are complex conjugate. The magnitudes of the two eigenvalues can be calculated with

$$|\lambda_{\mathbf{B}_i,1}|^2 = |\lambda_{\mathbf{B}_i,2}|^2 = \frac{1}{4}\Big((1+\bar{\beta}-\lambda_i)^2 - (1+\bar{\beta}-\lambda_i)^2 + 4\bar{\beta}\Big) = \bar{\beta}.$$

Define the imaginary part of $\lambda_{\mathbf{B}_i,1}$ and $\lambda_{\mathbf{B}_i,2}$ as $\Delta_i = -\delta(\lambda_i; \bar{\beta})$, that can be lower bounded by

$$\Delta_i \ge 4\bar{\beta} - (\bar{\beta} - \mu s + \mu)^2 > 0.$$

Concatenate the two eigenvectors together: $\mathbf{V}_i = [\mathbf{v}_{\mathbf{B}_i,1}, \mathbf{v}_{\mathbf{B}_i,2}]$ and the two eigenvalues: $\Lambda_{\mathbf{B}_i} = \begin{bmatrix} \lambda_{\mathbf{B}_i,1} & \\ & \lambda_{\mathbf{B}_i,2} \end{bmatrix}$. Then we have

$$\mathbf{B}_i = \mathbf{V}_i \Lambda_{\mathbf{B}_i} (\mathbf{V}_i)^{-1}$$

Upper bounds for $\|\mathbf{V}_i\|_2$ and $\|(\mathbf{V}_i)^{-1}\|_2$:

$$\|\mathbf{V}_i\|_2 = \left\| \begin{bmatrix} \lambda_{\mathbf{B}_i,1} & \lambda_{\mathbf{B}_i,2} \\ 1 & 1 \end{bmatrix} \right\|_2 \leq \sqrt{2+2\bar{\beta}}$$

$$\|(\mathbf{V}_i)^{-1}\|_2 \leq \frac{1}{|\lambda_{\mathbf{B}_i,1} - \lambda_{\mathbf{B}_i,2}|} \left\| \begin{bmatrix} 1 & -\lambda_{\mathbf{B}_i,2} \\ -1 & \lambda_{\mathbf{B}_i,1} \end{bmatrix} \right\|_2 \leq \frac{\sqrt{2+2\bar{\beta}}}{2(4\bar{\beta} - (\bar{\beta} - \mu s + \mu)^2)}$$

Concatenate the blocks together:

$$\mathbf{B} = \begin{bmatrix} \mathbf{B}_1 & & \\ & \cdots & \\ & & \mathbf{B}_{|S|} \end{bmatrix}, \Lambda_{\mathbf{M}} = \begin{bmatrix} \Lambda_{\mathbf{B}_1} & & \\ & \cdots & \\ & & \Lambda_{\mathbf{B}_{|S|}} \end{bmatrix}, \mathbf{V} = \begin{bmatrix} \mathbf{V}_1 & & \\ & \cdots & \\ & & \mathbf{V}_{|S|}. \end{bmatrix}$$

We have

$$\mathbf{M} = \underbrace{\begin{bmatrix} \mathbf{U} & \mathbf{0} \\ \mathbf{0} & \mathbf{U} \end{bmatrix} \mathbf{P} \mathbf{V}}_{\mathbf{T}} \Lambda_{\mathbf{M}} \underbrace{(\mathbf{V})^{-1} \mathbf{P}^{-1} \begin{bmatrix} \mathbf{U}^T & \mathbf{0} \\ \mathbf{0} & \mathbf{U}^T \end{bmatrix}}_{\mathbf{T}^{-1}}$$

and upper bounds for 2-norms of $\mathbf{T}$ and $\mathbf{T}^{-1}$:

$$\|\mathbf{T}\|_2 \leq \|\mathbf{V}\|_2 \leq \max_{i=1,2,\cdots,|S|} \|\mathbf{V}_i\|_2 \leq \sqrt{2+2\bar{\beta}},$$

$$\|\mathbf{T}^{-1}\|_2 \leq \|\mathbf{V}^{-1}\|_2 \leq \max_{i=1,2,\cdots,|S|} \|(\mathbf{V}_i)^{-1}\|_2 \leq \frac{\sqrt{2+2\bar{\beta}}}{2(4\bar{\beta} - (\bar{\beta} - \mu s + \mu)^2)}.$$

Claims (48) and (49) are proved. With the analysis in step 3, we finish the whole proof. □

## E   Proof of Theorem 2

To start the proof, we define some constants depending on $\mu, s, B, \underline{B}$:

$$C_1 := \max \left( C_0, \frac{1+\mu s - \mu}{1 - \mu s + \mu} \sqrt{s}B \right), \tag{53}$$

$$\widetilde{K}_1 := \left\lceil \frac{\log(\underline{B}) - \log(2C_1)}{\log(2\mu s - \mu)} \right\rceil + 2, \tag{54}$$

In the statement of the theorem, we assume "noise level $\sigma$ is small enough." Here we give the specific condition on $\sigma$:

$$\sigma \leq \min \left( \frac{B}{2}, \sqrt{s}B(2\mu s - \mu)^{\widetilde{K}_1} \right) \frac{1 - 2\mu s + \mu}{2sC_W}. \tag{55}$$

*Proof.* We first define parameters we choose in the proof:

$$\theta^{(k)} = \gamma^{(k)} \left( \mu \|\boldsymbol{x}^{(k)} - \boldsymbol{x}^*\|_1 + C_W \sigma \right), \tag{56}$$

$$p^{(k)} = \min \left( \frac{k}{\widetilde{K}_0} n, n \right), \tag{57}$$

The other two parameters $\gamma^{(k)}, \beta^{(k)}$ follow (34), (35) respectively as $k \leq \widetilde{K}_1$; $\gamma^{(k)}, \beta^{(k)}$ follow the step size and momentum rules in Conjugate Gradient (CG) respectively as $k > \widetilde{K}_1$:

$$\begin{aligned} \boldsymbol{r}^{(k)} &= (\boldsymbol{W}_{:,S})^T \left( \boldsymbol{A}_{:,S} \boldsymbol{x}_S^{(k)} - \boldsymbol{b} \right) \\ \alpha^{(k)} &= \|\boldsymbol{r}^{(k)}\|_2^2 / \|\boldsymbol{r}^{(k-1)}\|_2^2 \\ \boldsymbol{d}^{(k)} &= \begin{cases} \boldsymbol{r}^{(k)}, & k = \widetilde{K}_1 + 1 \\ \boldsymbol{r}^{(k)} + \alpha^{(k)} \boldsymbol{d}^{(k-1)}, & k > \widetilde{K}_1 + 1 \end{cases} \\ \gamma^{(k)} &= \|\boldsymbol{r}^{(k)}\|_2^2 / \left( (\boldsymbol{W}_{:,S} \boldsymbol{d}^{(k)})^T \boldsymbol{A}_{:,S} \boldsymbol{d}^{(k)} \right) \\ \beta^{(k)} &= \begin{cases} 0, & k = \widetilde{K}_1 + 1 \\ \alpha^{(k)} \gamma^{(k)} / \gamma^{(k-1)}, & k > \widetilde{K}_1 + 1. \end{cases} \end{aligned} \tag{58}$$

As $k \leq \widetilde{K}_1$, we follow the same proof line with Theorem 1 and obtain that $\boldsymbol{x}^{(k)}$ is close enough to $\boldsymbol{x}^*$ as $k = \widetilde{K}_1$ and it enters a small neighborhood of $\boldsymbol{x}^*$ where $\text{supp}(\boldsymbol{x}^{(k)}) = \text{supp}(\boldsymbol{x}^*)$. Moreover, the 'no false positive" conclusion also holds with the new instance-adaptive thresholds (56). Consequently, plugging (58) into (12), one can easily check that (12) is equivalent with applying a conjugate gradient (CG) method starting from $k = \widetilde{K}_1 + 1$ on the linear system

$$(\boldsymbol{W}_{:,S})^T \Big( \boldsymbol{A}_{:,S} \boldsymbol{x}_S - \boldsymbol{b} \Big) = \boldsymbol{0}. \tag{59}$$

We parameterize our system in a symmetric way (Section 3.1): $(\boldsymbol{W}_{:,S})^T \boldsymbol{A}_{:,S} = (\boldsymbol{D}^T \boldsymbol{D})_{S,S}$. And $(\boldsymbol{D}^T \boldsymbol{D})_{S,S}$ is positive definite according to (44) and $s < (1 + 1/\mu)/2$. Thus, the linear system (59) is a symmetric and positive definite system with solution $\overline{\boldsymbol{x}}_S$. It is well known that the conjugate gradient method on a symmetric and positive definite system has some important properties:

1. $\|\boldsymbol{x}_S^{(k)} - \overline{\boldsymbol{x}}_S\|_2$ is not monotone, but $\|\mathbf{D}_{:,S}(\boldsymbol{x}_S^{(k)} - \overline{\boldsymbol{x}}_S)\|_2$ is monotonic nonincreasing.

2. $\boldsymbol{x}_S^{(k)}$ stops at the solution in finite steps.

With the first property, we use the same techniques with those in Step 3 in the proof of Theorem 1 and obtain

$$\left\| \boldsymbol{x}_S^{(k)} - \overline{\boldsymbol{x}}_S \right\|_2 \leq \frac{1 + \mu s - \mu}{1 - \mu s + \mu} \left\| \boldsymbol{x}_S^{(\widetilde{K}_1)} - \overline{\boldsymbol{x}}_S \right\|_2, \quad \forall k > \widetilde{K}_1,$$

and $\boldsymbol{v}^{(k)} = \boldsymbol{0}$ for all $k > \widetilde{K}_1$. Consequently, $\text{supp}(\boldsymbol{x}^{(k)}) = \text{supp}(\boldsymbol{x}^*)$ holds for all $k > \widetilde{K}_1$.

The second property indicates that there is a large enough $\widetilde{K}_2$ such that $\boldsymbol{x}^{(k)} = \overline{\boldsymbol{x}}_S$ for all $k > \widetilde{K}_2$. In another word, there is a sequence $\{\overline{c}^{(k)}\}$ such that

$$\|\boldsymbol{x}^{(k)} - \overline{\boldsymbol{x}}_S\|_2 \leq \|\boldsymbol{x}^{(0)} - \overline{\boldsymbol{x}}_S\|_2 \prod_{t=0}^{k} \overline{c}^{(t)}$$

and it holds that

$$\overline{c}^{(k)} = 0, \quad \forall k > \widetilde{K}_2.$$

Finally, $\|\boldsymbol{x}^{(k)} - \boldsymbol{x}^*\| \leq \|\boldsymbol{x}^{(k)} - \overline{\boldsymbol{x}}_S\| + \|\overline{\boldsymbol{x}}_S - \boldsymbol{x}^*\|_2$. Combining the above upper bound and (46), we obtain (17). Theorem 2 is proved. $\qquad \square$

## F Instance-Adaptive Parameters for ALISTA without Momentum

Actually we can prove that ALISTA without momentum (setting $\beta^{(k)} = 0$ for all $k$ in (12)) also converge superlinearly if the parameters are taken in an instance-adaptive way. We list the theorem bellow and will discuss why we choose ALISTA-momentum rather than ALISTA without momentum following Theorem 3 and its proof.

**Theorem 3.** *Let $\boldsymbol{x}^{(0)} = \boldsymbol{0}$ and $\{\boldsymbol{x}^{(k)}\}_{k=1}^{\infty}$ be generated by (12) and set $\beta^{(k)} = 0$ for all $k$. If Assumption 1 holds and $s < (1 + 1/\mu)/2$ and $\sigma$ is small enough, then there exists a sequence of parameters $\{\theta^{(k)}, p^{(k)}, \gamma^{(k)}\}_{k=1}^{\infty}$ for each instance $(\boldsymbol{x}^*, \varepsilon) \in \mathcal{X}(B, \underline{B}, s, \sigma)$ so that we have:*

$$\|\boldsymbol{x}^{(k)} - \boldsymbol{x}^*\|_2 \leq \sqrt{s} B \prod_{t=0}^{k} \hat{c}^{(t)} + \frac{2sC_W}{1 - 2\mu s + \mu} \sigma, \quad \forall k = 1, 2, \cdots, \tag{60}$$

*where the convergence rate $\hat{c}^{(k)}$ satisfies*

$$\hat{c}^{(k)} \to 0 \ \text{ as } \ k \to \infty. \tag{61}$$

*Proof.* Similar with the previous two proofs, we introduce some constants to facilitate the proof.

$$\widetilde{K}_3 := \left\lceil \frac{\log(\underline{B}) - \log(2\sqrt{s}B)}{\log(2\mu s - \mu)} \right\rceil + 1. \tag{62}$$

We assume that
$$\sigma \le \min\left(\frac{B}{2}, \sqrt{s}B(2\mu s - \mu)^{\widetilde{K}_3}\right)\frac{1 - 2\mu s + \mu}{2sC_W}. \tag{63}$$

The parameters are taken as
$$\theta^{(k)} = \gamma^{(k)}\left(\mu\|\boldsymbol{x}^{(k)} - \boldsymbol{x}^*\|_1 + C_W\sigma\right), \quad \forall k \tag{64}$$

$$p^{(k)} = \min\left(\frac{k}{\widetilde{K}_3}n, n\right), \quad \forall k \tag{65}$$

$$\gamma^{(k)} = \begin{cases} 1, & k \le \widetilde{K}_3 \\ 1/\lambda_i, & k = \widetilde{K}_3 + i, 1 \le i \le |S| \\ \text{any real number}, & k \ge \widetilde{K}_3 + |S| \end{cases} \tag{66}$$

where $\lambda_i$ is the $i$-th singular value of $(\boldsymbol{W}_{:,S})^T\boldsymbol{A}_{:,S}$.

With (62), (63), (64), (65), (66), and following the same proof line as Theorem 1, one can prove that

- No false positive: $x_i^{(k)} = 0$ for all $i \in S$ and $k \ge 0$.

- Linear convergence: $\|\boldsymbol{x}^{(k)} - \boldsymbol{x}^*\|_2 \le \sqrt{s}B(2\mu s - \mu)^k + \frac{2sC_W}{1 - 2\mu s + \mu}\sigma$ for all $k \le \widetilde{K}_3$.

Since $\beta^{(k)} = 0$, the metric $\|\boldsymbol{x}^{(k)} - \boldsymbol{x}^*\|_2$ must be monotone and, consequently, $\boldsymbol{v}^{(k)} = \boldsymbol{0}$ for all $k \ge \widetilde{K}_3$ because $\boldsymbol{x}^{(k)}$ will stay in the neighborhood that $\text{supp}(\boldsymbol{x}^{(k)}) = \text{supp}(\boldsymbol{x}^*)$ once it enters the neighborhood. Thus, for $k > \widetilde{K}_3$, formula (12) reduces to
$$\boldsymbol{x}_S^{(k+1)} = \boldsymbol{x}_S^{(k)} - \gamma^{(k)}\left(\boldsymbol{W}_{:,S}\right)^T\left(\boldsymbol{A}_{:,S}\boldsymbol{x}_S^{(k)} - \boldsymbol{b}\right).$$

Denoting $\boldsymbol{Q} = (\boldsymbol{W}_{:,S})^T\boldsymbol{A}_{:,S}$, plugging in (45), we obtain
$$\boldsymbol{x}_S^{(k+1)} = \boldsymbol{x}_S^{(k)} - \gamma^{(k)}\boldsymbol{Q}(\boldsymbol{x}_S^{(k)} - \overline{\boldsymbol{x}}_S)$$
$$\boldsymbol{x}_S^{(k+1)} - \overline{\boldsymbol{x}}_S = \boldsymbol{x}_S^{(k)} - \overline{\boldsymbol{x}}_S - \gamma^{(k)}\boldsymbol{Q}(\boldsymbol{x}_S^{(k)} - \overline{\boldsymbol{x}}_S)$$
$$\boldsymbol{x}_S^{(k+1)} - \overline{\boldsymbol{x}}_S = \left(\boldsymbol{I} - \gamma^{(k)}\boldsymbol{Q}\right)(\boldsymbol{x}_S^{(k)} - \overline{\boldsymbol{x}}_S)$$

Since $\boldsymbol{Q}$ is real symmetric, it can be orthogonal diagonalized: $\boldsymbol{Q} = \boldsymbol{U}\Sigma\boldsymbol{U}^T$, where $\boldsymbol{U}$ is an orthogonal matrix and $\Sigma = \text{diag}(\sigma_1, \sigma_2, \cdots \sigma_n)$. Then,
$$\boldsymbol{x}_S^{(k+1)} - \overline{\boldsymbol{x}}_S = \left(\boldsymbol{U}\boldsymbol{U}^T - \gamma^{(k)}\boldsymbol{U}\Sigma\boldsymbol{U}^T\right)(\boldsymbol{x}_S^{(k)} - \overline{\boldsymbol{x}}_S)$$
$$\boldsymbol{U}^T(\boldsymbol{x}_S^{(k+1)} - \overline{\boldsymbol{x}}_S) = (\boldsymbol{I} - \gamma^{(k)}\Sigma)\boldsymbol{U}^T(\boldsymbol{x}_S^{(k)} - \overline{\boldsymbol{x}}_S)$$

Denoting $\boldsymbol{y}^{(k)} = \boldsymbol{U}^T(\boldsymbol{x}_S^{(k)} - \overline{\boldsymbol{x}}_S)$, we have
$$\boldsymbol{y}^{(k+1)} = (\boldsymbol{I} - \gamma^{(k)}\Sigma)\boldsymbol{y}^{(k)}.$$
$$y_i^{(k+1)} = (1 - \gamma^{(k)}\sigma_i)y_i^{(k)}, \quad i = 1, 2, \cdots, |S|.$$

Plugging in the parameters $\gamma^{(k)} = 1/\sigma^i$ for $k = \widetilde{K}_3 + i$, we have
$$y_1^{(k)} = 0, \quad \text{as } k \ge \widetilde{K}_3 + 2$$
$$y_2^{(k)} = 0, \quad \text{as } k \ge \widetilde{K}_3 + 3$$
$$\cdots$$
$$y_{|S|}^{(k)} = 0, \quad \text{as } k \ge \widetilde{K}_3 + |S| + 1$$

That is, $\boldsymbol{y}^{(k)} = \boldsymbol{0}$ for $k \ge \widetilde{K}_3 + |S| + 1$. Consequently, $\boldsymbol{x}^{(k)} = \overline{\boldsymbol{x}}$ for $k \ge \widetilde{K}_3 + |S| + 1$. This conclusion, combined with (46), implies (60) and (61) and finishes the proof. $\square$

Although both ALISTA and ALISTA-momentum acheives superlinear convergence, ALISTA-momentum performs better than ALISTA. To achieve superlinear convergence, ALISTA has to take step size as $\gamma^{(k)} = 1/\lambda_i$ (66) where $\lambda_i$ is one of the eigenvalues of $\boldsymbol{W}_{:,S}^T \boldsymbol{A}_{:,S}$. Such step size rule is not easy to calculate in practice and numerically unstable compared with the conjugate gradient in HyperLISTA. This is why we choose ALISTA-momentum and its instance-adaptive variant: HyperLISTA.

Theorem 3 itself shows that ALISTA with adaptive parameters converges faster than that with uniform parameters over instances. NA-ALISTA [4] learns instance-adaptive parameter rule with LSTMs and shows good numerical performance. Theorem 3 actually provides a theoretical explanation for this approach.

## G   Derivation of the Support Selection Formula

In this section, we provide the derivation of $p^{(k)}$ formula (24).

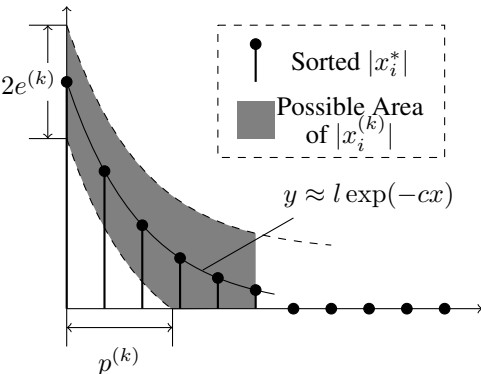

Figure 10: Choice of support selection (with more details)

We put the diagram of support selection (Figure 2) with more details in Figure 10. To estimate $p^{(k)}$, one should make assumptions on the distribution of the magnitudes of $x_i^*$. We assume that the nonzero entries of $\boldsymbol{x}^*$ follows normal distribution. The possibility of $x_i^*$ having a large magnitude is smaller than the possibility of $x_i^*$ having a small magnitude. Thus, the curve connecting "sorted $|x_i^*|$" in Figure 10 should be a convex function. Here we take exponential function $y = l \exp(-cx)$ to approximate the distribution. The intersection of the curve $y = l \exp(-cx)$ and the $y$-axis is $l = \max_i(x_i^*)$, i.e., $l = \|\boldsymbol{x}^*\|_\infty$. The lower bound of the gray area in Figure 10 is a shift of $y = l \exp(-cx)$. Thus, the relationship between $e^{(k)}, p^{(k)}$ can be formulated as

$$\|\boldsymbol{x}^*\|_\infty \exp(-cp^{(k)}) - e^{(k)} = 0.$$

Equivalently,

$$p^{(k)} = \frac{1}{c} \log \left( \frac{\|\boldsymbol{x}^*\|_\infty}{e^{(k)}} \right) = \frac{1}{c} \log \left( \frac{\|\boldsymbol{x}^*\|_\infty}{\|\boldsymbol{x}^{(k)} - \boldsymbol{x}^*\|_\infty} \right).$$

Constant $c$ depends on the distribution of $\boldsymbol{x}^*$. In practice, we use $c_3 = 1/c$ as the hyperparameter to tune. Furthermore, we use $\|\boldsymbol{A}^+ \boldsymbol{b}\|_1$ to estimate $\|\boldsymbol{x}^*\|$ and use residual $\|\boldsymbol{A}^+(\boldsymbol{A}\boldsymbol{x}^{(k)} - \boldsymbol{b})\|_1$ to estimate the error $\|\boldsymbol{x}^{(k)} - \boldsymbol{x}^*\|$. Then formula (25) is obtained. With an upper bound of $n$, (24) is obtained.