# OpenReview forum: "Hyperparameter Tuning is All You Need for LISTA"
_NeurIPS.cc/2021/Conference — NeurIPS 2021 Poster_

### Official Review · Reviewer_vByu · 2021-06-29

**Rating:** 6
**Confidence:** 3

**Summary:**

The authors propose an extension to the ALISTA algorithm. Momentum is added to the update rule to speed up convergence, and parameters are adaptively determined per layer, rather than learned over a training distribution. Such adaptivity was recently proposed in the NA-ALISTA algorithm as well, but this solution is more light-weight. HyperLISTA was shown to converge faster and to lower optimum than ALISTA (and variants thereof) in a synthetic data setup.

**Limitations And Societal Impact:**

The authors did not address limitations of their work. A discussion on improvements points/future research could be added in section 5.

**Main Review:**

The authors reduce the unfolded neural network implementation of ALISTA such that their algorithm finally reduces to having only 3 hyperparameters, that can be found by e.g. a grid search. The experiments on synthetic data show promising results. The main improvement point is the clarity of the explanation of the method itself.

**Originality**

The work is novel and cleverly combines aspects from different previous works, to find a more lightweight ALISTA algorithm.

**Quality**

L33: Reference [6] is not peer reviewed. More established (and peer-reviewed) papers are available that discuss the trade-off between model-based vs model-free methods.

L142: Does the convergence proof also hold for bounded noise, as assumed in the proof of ref 7? Otherwise it is unfair to compare convergence rates. Also in your experiments you do add noise.

From eq. 13 I understand that x^k+1 is updated more when the difference between x^k and x^k-1 is larger. I would expect opposite behavior, as you also suggest in l171: in low curvatures (so where the difference between  x^k and x^k-1 is small) you want a fast update.

L214: Why is it a logical choice in your opinion to use the L0 norm of x^k as a proxy for weighing the momentum factor?

L179: What proofs the claim you already make in this line?

Why did you choose 16 layers, since the errors clearly did not converge at 16, ALISTA-MM seems to become better than ALISTA-MM-Symm for more number of layers.

As you mentioned; you don’t need backpropagation anymore, but only have to find 3 hyperparameters. So why wouldn’t you do the grid search for these C1-C3 directly on new unseen (test data), rather than on training data? In that way you’d present you algorithm as just an iterative algorithm rather than an unfolded network, which includes a grid search on three settings (similar as many algorithms, e.g. ISTA, where you also need to find hyperparameters like thresholds and step sizes).

**Clarity**

One of the mentioned contributions is to add momentum to improve the convergence rate (also mentioned in L94). Can you be more specific already in the abstract on what entity you apply momentum?

L19: “each column of A, is normalized” -->  normalized to sum to 1?
And can you indicate here why this does help in simplicity, because it comes a bit out of the blue here.

L43-53: This paragraph partly contains information that is later again presented in the related work section. Also, lines 43 and 55 are exactly the same. The related work section seems a more detailed version of the mentioned paragraph, but written in a difference style (as if it was written by another author).

In L57 and L69, the authors speak about dictionary D, but in equation 1 the dictionary was called A.

L91: “First, we start with a new symmetric parameterization for the weight matrix in ALISTA so that ALISTA also converges to LASSO function minimizers, that lays the foundation of momentum-based analysis.”
This sentence contains a lot of information that has not been presented before. The reader is not aware yet about the non-symmetric parameterization of the weight matrix in ALISTA, nor about any momentum-based analyses.

L91-98: In the abstract three main contributions are mentioned, while in lines 91-98 four contributions are mentioned. The first of these four however comes out of the blue.

*Section 2*

To improve readability and structure of this section, it would be wise to add subsection headers named 'Setup and Notations', 'ISTA', 'LISTA', 'ALISTA', 'Assumption 1'. It is nice for completeness that your also provide the background on ISTA, but having it separated with subsections makes it easier for the reader when you want to search back the ALISTA notation later on during reading the paper.

L137: “In ALISTA, p(k) is regarded as a hyperparameter to be manually tuned by cross validation or so.” --> Is it typically tuned separately per index k in ALISTA, so generating K hyperparameters, or is it typically set to the same value for all layers?

L140: From eq. 8 I understand that the elements in x* are strictly larger than 0, since 0 < B_ < x*_i. But the definition of sparse elements is that they are exactly zero right?

*Section 3*

L146: Please add a short introduction about the section. Mention that you will extend the ALISTA framework (as provided in a separate subsection in related work, so that the reader can easily refer back to those formulas), and which aspects you will improve upon. Because now I suddenly jump into the title Symmetric Jacobian of Gradients which seems out of the blue, when coming from the related work section, and L148 only implicitly says that you will continue upon formula 5.

*Section 3.1*

I have most problems understanding section 3.1. Here are some tips/questions that might improve its clarity:

L 148: It was not explained why good weights should have low coherence with matrix A, and why W^TA should be symmetric, which is in fact the key take away of this section.

L 148: It is unclear to me how the two problems formulated in l148 (i.e. non-symmetry and no gradient of a (which?) function)  hamper convergence to F(x).

L151: Here you mention that ALISTA can only be solved using the ground truth, while from equation 5 I understood something different. Why can’t you use eq. 5 directly, instead of resorting to the method by ref. 1 using gram matrices?

L161: What do you mean by “it usually holds in practice”. This method using the Gram matrix is proposed by yourself, so which practice are you referring to? The current paper or earlier work?

L163: “Such conclusion demonstrates that procedures (10) and (5) lead to similar mutual coherence and, consequently, one can also pursue x(k) -> x* with the new dictionary GA, with almost no performance loss.”
--> Which conclusion do you refer to, mutual coherence of what are you referring to,  and which performance?

L164: If indeed it doesn’t matter much whether we use the symmetric or non-symmetric dictionary, why would we then do all the effort at all to find this symmetric dictionary D?

*Section 3.2*

L167: “… the above new symmetric parameterization make us be able to unify the proof schemes in ALISTA and Polyak’s heavy ball momentum, which was not feasible previously.”  --> The Polyak’s heavy ball momentum was not introduced before, and the sentence seems grammatically incorrect as well.

L213: You write that you estimate ||x^k -x*||, but in eq. 17 this term is different: ||x^k(x*)-x*||.


L238: “In practice, we choose the switching condition as p(k) large enough.” --> I don’t understand what you mean by choosing p(k), as p(k) is adaptively computed, rather than chosen. So how can you guarantee that it is ‘large enough’.

From section 3.5 I understand that HyperLISTA in the end is not trained with backpropagation, so why would you then still implement it as an unfolded neural network and not just as an iterative algorithm? Is it still accelerated as an unfolded NN compared to using it as an iterative algorithm maybe?

A textual reference to figure 1 is missing, and a more informative stand-alone caption could be useful.

*Algorithmic description*

After having read the different formulas, it is a pleasure to see a summarizing algorithmic description. However, the description is not stand-alone (enough). A reader that does not want to be bothered with the proofs is not able to follow it like a recipe.
Looking at the equations needed in the algorithm, it seems that next to **b**, also **A** must be given. Moreover, given **A**, the user should solve eq. 10 to find D, in order to be able to compute mu, as used in eq. 21 and eq. 23. This information is missing from the description.

*Section 4*

Fig. 3: Introduce the meaning of the abbreviations in the legend in section 4.1.

Fig. 4: What is the difference in implementation between HyperLISTA-GS and HyperLISTA-Full, this is not clear from paragraph 4.2.
Why is comparison to NA-ALISTA only added in section 4.3, but not in 4.1 and 4.2?


**Significance**
Reducing the number of hyperparameters in existing algorithms is an interesting direction, and of use to the community.

**Minor points**

L39: double ‘for’ in the sentence.

L115: represent instead of represents

L121: You mention eta_theta while in the formula you use eta_lambda/L.

L 126: Please mention that x^* is the optimal solution (i.e. the ground truth). It’s now implicit from eq. 1, but I was initially confused, since x^* was used in eq.1, but not in eq.2, and again in eq. 3.

L133: It looks like your forgot bold notation of the last v in this line.

L142: In this line your write B_ < x*_i, while in eq. 8 your use a smaller-equal sign for that inequality.

L153: The set of reals is here indicated with a different R than used in the preceding text.

L157: adopt --> adopt

L159: The gradient with respect to which variable?

L186: How should I interpret this notation: x^k(x*). Is it a dot product or a function operator maybe?

L229: The notation in the denominator of the second term seems faulty.


**Time Spent Reviewing:**

5

---

> ### Author Response · Authors · 2021-08-10
> **Part 3. Responses to the comments on “Minor Points” and others**
>
> * Minor points
>
> Thanks for such detailed comments! To clarify the reviewer’s confusion, we list some questions and our responses below. Other comments are typos, we will fix them as your suggestion in our final paper.
>
> * L121: You mention eta_theta while in the formula you use eta_lambda/L.
>
> Sorry for the confusion. Actually, $\eta_\theta$ is a general operator defined for all $\theta$. In the formula, we use $\theta = \lambda/L$. We will clarify this point in the revision.
>
> * L 126: Please mention that x^* is the optimal solution (i.e. the ground truth). It’s now implicit from eq. 1, but I was initially confused, since x^* was used in eq.1, but not in eq.2, and again in eq. 3.
>
> We will mention $x^*$ is the optimal solution in our final paper.
> To address the confusion, eq. (2) is an iterative formula to be applied in sparse signal recovery. When we apply the formula (2), we don’t know the ground truth. The ground truth is the target of (2).
> However, eq. (3) is a training loss function to train the parameters in (2). In supervised learning, the ground truth may be known and used in training.
>
> * L142: In this line your write B_ < x*_i, while in eq. 8 your use a smaller-equal sign for that inequality.
>
> It is smaller-equal, we will fix the typo.
>
> * L159: The gradient with respect to which variable?
> Gradient with respect to x. We will add this point.
>
> * L186: How should I interpret this notation: x^k(x*). Is it a dot product or a function operator maybe?
>
> It is a function operator. We will define this notion in Section 2.
>
> ---
>
> * Limitations And Societal Impact:
>
> * The authors did not address limitations of their work. A discussion on improvements points/future research could be added in section 5.
>
> Thanks for pointing this out. A dictionary with perturbation is not considered in this work now. One may follow the “robust ALISTA” method in [14] to improve HyperLISTA in the future.

---

> > ### Comment · Reviewer_vByu · 2021-08-17
> > **Response to author's reply**
> >
> > Dear authors,
> >
> > Thank you for your great efforts in promptly answering all my concerns.
> > In general I'm satisfied with your detailed answers. Nevertheless, I'll be waiting for a revised version to check readability and clarity, to decide upon increasing the grade.
> >
> > Regarding the last comment about limitations.
> > In line with a comments from reviewer WJ5N, I think a bit more discussion might still be useful for the reader. You now mention one technical suggestion for future work (in your reply), but discuss for example things that you have tried but did not work, or challenges you encountered  in this algorithm. It may help future researchers that aim to use the proposed algorithm.

---

> > > ### Author Response · Authors · 2021-08-17
> > > **Thank you! And regarding the "revised version"**
> > >
> > > Dear Reviewer vByu:
> > >
> > > Thank you so much for appreciating our rebuttal! We are very delighted that your technical concerns now seem to have been fully resolved; and that our reviewers agree on this paper's technical significance, novelty, and originality.
> > >
> > > Some discussion of future improvement points, such as robustness to dictionary perturbations (technical suggestion), or its uninsured performance when the noise distribution changes (challenge encountered), can also be easily appended to the revised paper as a short paragraph in Section 5. We will happily include them.
> > >
> > > Regarding your request of a "revised version", we are however uncertain what we should do at the moment since **NeurIPS rebuttal policy** officially stated:
> > > * No paper revisions may be submitted during the review process.
> > >
> > > Based on our above point-to-point responses (most lead to minor clarifications that are very easy to include), we are obviously ready to present a revised version quickly. We actually have already started revising at our end. But due to the above policy, there is no way for us to directly upload a revised version to OpenReview.
> > >
> > > We are considering the following alternatives, and please let us know your preference or recommendation:
> > > * Option 1: if there is any particular part(s) of paper that you would like to see us to revise, in order to decide whether it meets your clarity bar: could you please specify such part(s)? We are happy to paste their revisions as plain text, in the response
> > > * Option 2: or, if the area chair will approve so, we could use an anonymous link (removing all identifying information that may violate the double-blind reviewing policy), to the revised draft as a linked PDF. But we are really uncertain whether this will go against the NeurIPS rebuttal policy, and we hope AC could give the guidance here on whether this is appropriate.
> > >
> > > Since the "readability and clarity" of our revised final paper seems to be your last remaining question, we are willing to do whatever is allowed to address that. Again, we are very thankful for your valuable time and suggestions.

---

> > > > ### Comment · Reviewer_vByu · 2021-08-25
> > > > **Reply to author's reply**
> > > >
> > > >
> > > > Dear authors,
> > > >
> > > > Sorry for coming back to you with quite a delay. I re-read your reply another time, and given that I now know that you cannot upload a revised version, I’ve some last minor points:
> > > >
> > > > 1.	I understand the answer you gave on this question:
> > > > “Fig. 4: What is the difference in implementation between HyperLISTA-GS and HyperLISTA-Full, this is not clear from paragraph 4.2. Why is comparison to NA-ALISTA only added in section 4.3, but not in 4.1 and 4.2?”
> > > > However, you did not mention that you will add this to the manuscript as well. I think it is still useful to also inform the reader about the exact difference between HyperLISTA-Full and HyperLISTA-GS. So are you planning to clarify this a bit better in the paper as well?
> > > >
> > > > 2.	I think my last concern (in the reply) was only partly answered. You indeed indicate that you can add some points regarding future improvements. I think the paper would even be more improved/fair if you also report which things did not work out so well or which limitations you found during your experiments. This might help people that aim to use the algorithm. For example, now you have added also an experiment on real images; did you find issues that were not present with the synthetic data? And can you say something about the scope of real-life data for which this algorithm is unsuitable, in order to correctly indicate the boundaries of the algorithm?
> > > >
> > > > Further than that, I think the algorithmic description is much better the way you proposed to do it now.
> > > > Lastly, after making all the changes I would recommend re-reading the full paper again (or maybe let it do by someone else with a fresh view) to see whether readability improved and whether a new reader can better follow the story line now. Based upon the answers that you provided, I belief the readability is much better now, thanks to the added subsection titles and the removal of several sentences that came too early in the story.
> > > >
> > > > I'm intending to increase your grade, of course awaiting your final reply.
> > > > Congrats with the great rebuttal!

---

> > > > > ### Author Response · Authors · 2021-08-25
> > > > > **Response to the follow-up questions**
> > > > >
> > > > > Dear Reviewer vByu,
> > > > >
> > > > > Thank you for your reply and discussion that helped improve the quality of this manuscript. Upon you follow-up questions:
> > > > >
> > > > > 1. Yes, we promise to add our explanation in the rebuttal to the manuscript as well. We intended to carefully integrate everything in our rebuttal and rolling discussions into the manuscript, to ensure that our final version is clear in every way. We would also like to thank you again for your detailed feedback that points out the unclear parts in our manuscript.
> > > > >
> > > > > 2. Our last response has mentioned a practical limitation in considering the perturbed or inaccurate dictionary. Now, more specifically to your latest inquiry regarding in “real image experiments'', we have found that if the test images have complicated structures that are beyond the assumptions in our paper, then both ALISTA (ICLR 2019) and HyperLISTA may fail to perform well, because their underlying sparse linear model in eq. 1 becomes oversimplified and no longer holds well. In this case, models with larger learning capability such as ISTA-Net (CVPR 2018) will be better than the compact ALISTA and HyperLISTA models. We believe the complexity and availability of training data will draw a boundary of the ALISTA/HyperLISTA algorithms, and we will mention this point clearly in the revised paper.
> > > > >
> > > > > We would like to express our gratitude for your appreciation in our efforts to improve the clarity and readability. We will make sure all the mentioned points are implemented in the final version. We will also ask other peers to proofread the revision.
> > > > >
> > > > > Thank you for your valuable time and constructive comments!
> > > > >
> > > > > Best wishes,
> > > > >
> > > > > Paper8022 Authors

---

> > > > > > ### Comment · Reviewer_vByu · 2021-08-31
> > > > > > **Upload revised version**
> > > > > >
> > > > > > Dear authors,
> > > > > >
> > > > > > The area chair has told me that he/she allowed you to upload a revised version.
> > > > > > Since this is possible now, I'm waiting for that version upon deciding a final grade.
> > > > > >
> > > > > > Kind regards

---

> > > > > > > ### Comment · Reviewer_vByu · 2021-09-01
> > > > > > > **Review revised version**
> > > > > > >
> > > > > > >
> > > > > > > Dear authors,
> > > > > > >
> > > > > > > I’ve run another check over the revised manuscript. I think many things have already been improved since the first submission. However, I’ve still some remarks about points that remain a bit unpolished:
> > > > > > >
> > > > > > > L105: You now added the comment that X^(K)(\Theta,b,x^0) is a function. However, in your first reply to me you mentioned that you omitted these parameters in the rest of the paper for readability reasons. This is still not mentioned in the paper, and it is therefore still confusing that you use two different notations. Besides,  just two lines above eq. 3, x^(k+1) is defined as a vector rather than a function, so eq. 2 and 3 are inconsistent.
> > > > > > >
> > > > > > > L138: “remove its main concepts”; do you truly mean to say remove here?
> > > > > > >
> > > > > > > L146: Typo: one has (instead of have)
> > > > > > >
> > > > > > > Figs 3-7: Tip: the figures would be easier interpretable, if you would in all figures use the same color for the same model.
> > > > > > >
> > > > > > > L326: “We also find that HyperLISTA, as a compact model similar to ALISTA [17], may fail to perform well on signals with complicated structures such as real images, where the underlying sparse linear model (1) becomes oversimplified and no longer holds well. We will investigate these further in the future.”
> > > > > > > --> I indeed asked you to mention shortcomings of the current work, but the thing mentioned here now is not in line with the added results presented in appendix B, where HyperLISTA outperformed ALISTA. Or were the results visually pretty bad still? If that’s the case, and you want to make that point here, it would be fair to add the visual results as well in the appendix.
> > > > > > >
> > > > > > > Given the fact that the paper still contains typos and some unclarities, after heavy updates, I share the opinion of reviewer NZWV and would advice an additional round of polishment before submitting it again somewhere else to receive fresh reviews.

---

> > > > > > > > ### Author Response · Authors · 2021-09-02
> > > > > > > > **Response to your comments on the revised pdf**
> > > > > > > >
> > > > > > > > Dear Reviewer vByu,
> > > > > > > >
> > > > > > > > Thank you for your timely response as to the revised pdf. You included five new comments. Two of them (L138 and L146) are minor typos. We apologize for that and have already fixed them (see the updated pdf in the new link https://tinyurl.com/spyueax6). Another comment is about the color usage in figures for better illustrations. Thank you for the great suggestion and we also integrated it into the updated pdf.
> > > > > > > >
> > > > > > > > Upon the other two comments in your reply, we have our responses below.
> > > > > > > >
> > > > > > > > **Notation in L105.**
> > > > > > > >
> > > > > > > > We decided to still add that notation in eqn. 3 because we think it will be confusing to omit \Theta while we are optimizing over \Theta and also because the network output X^(K) actually depends on \Theta. To address your concern, we further added in the updated pdf that “We will use x^(K) to refer to x^(K)(\Theta, b, x^(0)) for brevity in the remainder of this paper.”
> > > > > > > >
> > > > > > > > However, we humbly argue that our notations in eqn. 2 and 3 do have a coherent and clear logical flow and we are actually following the well-accepted conventions in peer-reviewed papers [6,29,33]. A LISTA model can be described as a neural network if we consider x^(k) as hidden variables and formulate what happens in each layer in eqn. 2, OR be described as a cascaded function where x^(k) are the intermediate results as we formulated in eqn. 3. Such interchangeable notations are widely used in the literature and we believe that they will cause no confusion.
> > > > > > > >
> > > > > > > > **Clarification on L326**
> > > > > > > >
> > > > > > > > *Firstly*, the visual quality of recovered images is not bad compared to Recon-Net (CVPR 2016). We provided a few visual examples in Appendix B. Please feel free to check the visual results in the updated pdf.
> > > > > > > >
> > > > > > > > *Secondly*, we would like to kindly point out that we were not comparing HyperLISTA against ALISTA by saying “We also find that HyperLISTA, as a compact model similar to ALISTA [17], may fail to perform well on...”. On the contrary, what we originally meant here is that **BOTH** ALISTA and HyperLISTA belong to theory-grounded “compact” LISTA models. In contrast, another branch of empirical overparameterized LISTA variants, such as ISTA-Net (CVPR’18) that we mentioned in our previous response to you, is not accompanied with theoretical results, but might work empirically better on complicated images.
> > > > > > > >
> > > > > > > > ---
> > > > > > > >
> > > > > > > > Again, we express our sincere gratitude for your active involvement in the rolling discussions and valuable comments that helped us improve our work.
> > > > > > > >
> > > > > > > > Best,
> > > > > > > >
> > > > > > > > Paper8022 Authors

---

> ### Author Response · Authors · 2021-08-10
> **Part 2. Responses to the comments on “Clarity” (Sections 3.2 ~ 4)**
>
> * Section 3.2
>
> * L167: “… the above new symmetric parameterization make us be able to unify the proof schemes in ALISTA and Polyak’s heavy ball momentum, which was not feasible previously.” --> The Polyak’s heavy ball momentum was not introduced before, and the sentence seems grammatically incorrect as well.
>
> Thanks for pointing this out. This sentence is to describe the connection between Sections 3.1 and 3.2. As the reviewer suggested, we add a short introduction paragraph in L146 (between Sections 3 and 3.1), then the sentence in L167 here is not necessary. We will remove this sentence in our next version .
>
> * L213: You write that you estimate ||x^k -x*||, but in eq. 17 this term is different: ||x^k(x*)-x*||.
>
> We use the notation x^k(x*) because x^k is a function of the input, i.e., the measurement b, which is decided by the ground truth signal x^* by the generative model b = A x^* (assuming no measurement noise).\
> In our next version, bounded measurement noise is considered. “b=Ax^*+epsilon.” Thus, x^k depends on x^* and epsilon. Then the notion will be x^k(x^*, epsilon). For simplicity, we write x^k to make the notation not too wordy. We will clarify this point in Section 2.
>
> * L238: “In practice, we choose the switching condition as p(k) large enough.” --> I don’t understand what you mean by choosing p(k), as p(k) is adaptively computed, rather than chosen. So how can you guarantee that it is ‘large enough’.
>
> Thanks for your comment. You are correct that p(k) is adaptively computed. We mean to choose the switching condition based on the computed p(k). Based on (24), p(k) increases as k increases. As k = 0, we use the update rule of ALISTA-MM (13); as k increases, we keep using (13); once k is large enough such that p(k)>= threshold, we switch to the conjugate gradient. \
> In empirical experiments, we find that it is safe to switch when p(k) is over (0.2 * n), i.e., over 20% of the coordinates in x^(k) are selected by the support selection. We will clarify this in the revised version.
>
> * From section 3.5 I understand that HyperLISTA in the end is not trained with backpropagation, so why would you then still implement it as an unfolded neural network and not just as an iterative algorithm? Is it still accelerated as an unfolded NN compared to using it as an iterative algorithm maybe?
>
> We agree with the reviewer that HyperLISTA can be viewed as an iterative algorithm because it only has 3 hyperparameters to tune. We may either implement HyperLISTA as a neural network or as an iterative algorithm:
> 1. As an unfolded neural network, HyperLISTA is “trained” on a training set. The training is actually hyperparameter optimization. Once we obtain the 3 “trained” hyperparameters, we apply them directly to the testing set.
> 2. As an iterative algorithm, HyperLISTA can be viewed as hyperparameter optimization for each testing dataset, to obtain superlinear convergence. We will add in revision.
>
> * A textual reference to figure 1 is missing, and a more informative stand-alone caption could be useful.
>
> Thank you for pointing this out. Figure 1 is an illustration of the iterations in ALISTA. We will add the textual reference and necessary information to the caption in the revision.
>
> * Algorithmic description
>
> * After having read the different formulas, it is a pleasure to see a summarizing algorithmic description. However, the description is not stand-alone (enough). A reader that does not want to be bothered with the proofs is not able to follow it like a recipe. Looking at the equations needed in the algorithm, it seems that next to b, also A must be given. Moreover, given A, the user should solve eq. 10 to find D, in order to be able to compute mu, as used in eq. 21 and eq. 23. This information is missing from the description.
>
> Thank you for pointing this out. We will present algorithm 1 in the revision like this:
>
> 1. Input: observation b, measurement matrix A, hyperparameters $c_1,c_2,c_3$.
> 2. Initialize: set $x^{(0)} = 0$.
> 3. Calculate $D$ and $G$ with (10), set $W = (G^TG)A$.
> 4. Calculate $\mu$ with $\mu = \max_{i \neq j} | D_{:,i}^TD_{:,j} | $
> 5. For loop (the remaining part keeps the same as the current draft)
>
> ---
>
> * Section 4
>
> * Fig. 3: Introduce the meaning of the abbreviations in the legend in section 4.1.
>
> Thank you for the suggestion. We will add the meaning of the abbreviations in Section 4.1. We will also add another paragraph at the beginning of Section 4 describing the terminology of the models used and compared in the experiments.
>
> * Fig. 4: What is the difference in implementation between HyperLISTA-GS and HyperLISTA-Full, this is not clear from paragraph 4.2. Why is comparison to NA-ALISTA only added in section 4.3, but not in 4.1 and 4.2?
>
> In short, HyperLISTA-GS only optimizes the hyperparameter c_1 and c_2 while HyperLISTA-Full also optimizes c_3. The motivation for the experiment design is that HyperLISTA is the counterpart of HyperLISTA-BP, where c_1 and c_2 are trained using back-propagation (SGD). c_3 is not trained in HyperLISTA-BP because c_3 decides the ratio of coordinates selected by support selection, which is not differentiable. For fairness, we only optimize c_1 and c_2 when comparing HyperLISTA-BP and HyperLISTA-GS. In contrast, HyperLISTA-Full is the ultimate version of HyperLISTA where we search all three hyperparameters.
>
> * Significance Reducing the number of hyperparameters in existing algorithms is an interesting direction, and of use to the community.
>
> Thanks for your encouragement.

---

> ### Author Response · Authors · 2021-08-10
> **Part 2. Responses to the comments on “Clarity” (Sections 1~3.1)**
>
> * One of the mentioned contributions is to add momentum to improve the convergence rate (also mentioned in L94). Can you be more specific already in the abstract on what entity you apply momentum?
>
> Yes, in the abstract, we rewrite: “we show that adding momentum to intermediate variables in the LISTA network achieves a better convergence rate”
>
> * L19: “each column of A, is normalized” --> normalized to sum to 1? And can you indicate here why this does help in simplicity, because it comes a bit out of the blue here.
>
> Yes, we mean normalizing the columns of dictionary $A$ to have unit l-2 norms. The motivation for doing so is that scaling the columns of $A$ has a direct influence on the solution of the LASSO optimization problem (formulated in the second paragraph in Section 2). Normalizing the columns in $A$ is a good way to normalize the solution of the LASSO problem. This is also a well-adopted normalization technique in the literature [1,2].
>
> [1] Donoho & Elad, 2003. “Optimally sparse representation in general (nonorthogonal) dictionaries via l1 minimization.” PNAS.\
> [2] Candes, Romberg, & Tao, 2006. “Stable Signal Recovery from Incomplete and Inaccurate Measurements.”
>
> * L43-53: This paragraph partly contains information that is later again presented in the related work section. Also, lines 43 and 55 are exactly the same. The related work section seems a more detailed version of the mentioned paragraph, but written in a difference style.
>
> Thank you for the comments. We will remove this part and remain the related work section.
>
> * In L57 and L69, the authors speak about dictionary D, but in equation 1 the dictionary was called A.
>
> Thank you for pointing this out. We will fix this typo and make sure that all notations are consistent in the revision.
>
> * L91: “First, we start with a new symmetric parameterization for the weight matrix in ALISTA so that ALISTA also converges to LASSO function minimizers, that lays the foundation of momentum-based analysis.” This sentence contains a lot of information that has not been presented before. The reader is not aware yet about the non-symmetric parameterization of the weight matrix in ALISTA, nor about any momentum-based analyses.
>
> Thank you for the comment. We will remove this sentence from the contribution part since it is just a preliminary part to lay the foundation of momentum-based analysis.
>
> * L91-98: In the abstract three main contributions are mentioned, while in lines 91-98 four contributions are mentioned. The first of these four however comes out of the blue.
>
> We will remove the first point in L91. (discussed above)
>
> ---
>
> * Section 2
>
> * To improve readability and structure of this section, it would be wise to add subsection headers named 'Setup and Notations', 'ISTA', 'LISTA', 'ALISTA', 'Assumption 1'. ........
>
> Thank you for the great suggestion! We will definitely add such a subsection including all those information and context in the revised version.
>
> * L137: “In ALISTA, p(k) is regarded as a hyperparameter to be manually tuned by cross validation or so.” --> Is it typically tuned separately per index k in ALISTA, so generating K hyperparameters, or is it typically set to the same value for all layers?
>
> According to the original ALISTA paper [14], $p(k)$ is proportional to the index $k$, capped by a maximal value, i.e., $p(k) = \min(p*k, p_{max})$, where $p$ and $p_{max}$ are two hyperparameters that will be tuned manually.
>
> * L140: From eq. 8 I understand that the elements in x* are strictly larger than 0, since 0 < B_ < x*_i. But the definition of sparse elements is that they are exactly zero right?
>
> Thanks for pointing out this typo. We mean that the magnitudes, or the absolute values, of the  NON-ZERO elements in x^* are strictly larger than 0. It should be written as \
> $0 < B_ \leq |x*_i| \leq B, \forall i \in \mathrm{support}(x^*)$.\
> And the number of non-zero elements is bounded by s. All other elements are strictly zero.
>
> ---
>
> * Section 3
>
> * L146: Please add a short introduction about the section. Mention that you will extend the ALISTA framework (as provided in a separate subsection in related work, so that the reader can easily refer back to those formulas), and which aspects you will improve upon. Because now I suddenly jump into the title Symmetric Jacobian of Gradients which seems out of the blue, when coming from the related work section, and L148 only implicitly says that you will continue upon formula 5.
>
> We will add a short introduction here: “In this section, we extend the ALISTA framework (eq. 4) in three aspects: (1) we introduce a better method to calculate matrix W in formula (eq. 5); (2) we augment the x-update formula (eq. 4) by adding a momentum term; (3) we propose a novel approach to obtain the parameters $p^{(k)}, \theta^{(k)}, \gamma^{(k)}$ in (eq. 4).”
>
> ---
>
> * Section 3.1
>
> * L 148: It was not explained why good weights should have low coherence with matrix A, and why W^TA should be symmetric, which is in fact the key take away of this section.
>
> Mutual coherence is a key concept in compressive sensing introduced by (Donoho & Elad, 2003). It characterizes the cross-correlations between the columns of $A$. A matrix with low mutual coherence satisfies that $A^TA$ is approximately equal to the identity matrix. In another word, operator $A$ is close to the full sampling of the sparse vector $x^*$, thus implying a high probability of an accurate reconstruction from the measurement $b$. In ALISTA [14], the authors extend this concept to the coherence between matrix $A$ and another matrix $W$. By minimizing the coherence between $W$ and $A$, they obtained a good matrix $W$ that can be plugged into the LISTA framework.
>
> One limitation of ALISTA is that ALISTA is only proved to converge to $x^*$. It is shown in Step-LISTA [3] that with the default parameterization (eq. 5), ALISTA cannot converge to the LASSO minimizer. This is partially due to the fact that the update direction $W^T(Ax - b)$ in ALISTA is not aligned with the gradient of the l2 term in the LASSO objective, $\nabla_x ½ ||Ax-b||^2_2 = A^T(b-Ax)$. This limitation motivates us to introduce symmetric Jacobian parameterization. With the new parameterization, the LASSO objective can be reformulated as in eq. 11. In this case, the new update direction in ALISTA-Symm will be aligned with the gradient of $½ ||G(Ax-b)||^2_2$.
>
> * L 148: It is unclear to me how the two problems formulated in l148 (i.e. non-symmetry and no gradient of a (which?) function) hamper convergence to F(x).
>
> Sorry for the ambiguity of the “function” here. We meant the l2 term in the LASSO objective, i.e., $\nabla_x ½ ||Ax-b||^2_2 = A^T(b-Ax)$. It is shown in the Step-LISTA paper [3] that the non-symmetric parameterization in ALISTA does not converge to the minimizer of $F(x)$. An intuitive explanation is that the update direction $W^T(b-Ax)$ is not aligned with the gradient $A^T(b-Ax)$ mentioned above. We will make this part clearer in the revised version.
>
> * L151: Here you mention that ALISTA can only be solved using the ground truth, while from equation 5 I understood something different. Why can’t you use eq. 5 directly, instead of resorting to the method by ref. 1 using gram matrices?
>
> ALISTA actually uses the eq. 5 to solve the coherence minimization problem to get the weight matrix $W$. For the reason mentioned above, this weight matrix $W$ can not provide convergence to the LASSO minimizers. Therefore, we resort to the symmetric parameterization in Ada-LISTA [1].
>
> * L161: What do you mean by “it usually holds in practice”. This method using the Gram matrix is proposed by yourself, so which practice are you referring to? The current paper or earlier work?
>
> We mean that we find the approximated equality usually holds in our empirical experiments. We will rewrite this sentence as “it usually holds that”.
>
> * L163: “Such conclusion demonstrates that procedures (10) and (5) lead to similar mutual coherence and, consequently, one can also pursue x(k) -> x* with the new dictionary GA, with almost no performance loss.” --> Which conclusion do you refer to, mutual coherence of what are you referring to, and which performance?
>
> 1. Which conclusion: we refer to conclusion (12)
> 2. Mutual coherence of what: “the mutual coherence between A and (G^TG)A obtained by (10)” and “the mutual coherence between A and 3. W obtained by (5)” are similar.
> 4. Which performance: if we plug the matrices obtained by (10) into the framework of ALISTA, the recovery performance almost has no loss compared with plugging (5) into ALISTA.
>
> We will rephrase the sentence as: “Conclusion (12) demonstrates that the mutual coherence between $A$ and $(G^TG)A$ obtained by (10) is similar to that between A and W obtained by (5). Although (10) limits the matrix W^TA to be symmetric, it performs almost without loss in the framework of ALISTA compared with (5).”
>
> * L164: If indeed it doesn’t matter much whether we use the symmetric or non-symmetric dictionary, why would we then do all the effort at all to find this symmetric dictionary D?
>
> Symmetric Jacobian has two merits:
> 1. Theorems 1 and 2 depend on the symmetric dictionary. The symmetric real matrix is orthogonally diagonalizable. We need this property to establish the convergence of ALISTA-MM in Theorem 1. Theorem 2 depends on the convergence proof of the conjugate gradient. The conjugate gradient method converges superlinearly only as the linear system is symmetric.
> 2. Meanwhile, the symmetric jacobian provides the flexibility to choose loss function in training. If we have the ground truth x^*, we can train the model with loss function $||x^K- x^*||^2$. If we do not have the ground truth, we can train the model with loss function:
> $½ ||G(Ax-b)||^2_2 + \lambda * ||x||_1$. This allows us to train ALISTA in an unsupervised approach.

---

> ### Author Response · Authors · 2021-08-10
> **Part 1. Responses to the comments on “Quality”**
>
> Dear reviewer, thank you for the detailed comments on our work, which help very much for us to identify the unclarity in the current presentation and significantly improve the overall writing quality and readability.
>
> --------------------------------------------------------------------------------------------------------------------------------------------------
>
> * Reference [6] in L33.
>
> We follow the definitions in [6], the latest review article released in this field. To our best knowledge, [6] was the first to propose those terms. Since this is not yet peer-reviewed, we are happy to remove those terms from the introduction without affecting any main point made in this paper, if the reviewer suggests so. We are also happy if the reviewer may provide some peer-reviewed papers discussing this point.
>
> ----------------------------------------------------------
>
> * Convergence of HyperLISTA with the presence of bounded noise.
>
> Yes, we are able to extend our convergence proof to the case with bounded observation noise. In this case, the recovery error cannot converge to zero due to the noise. However, we can prove that our eventual recovery error is no worse than [7], and our convergence rate is exactly faster than [7]. This is consistent with the current Theorems 1 & 2 in the paper.
>
> Our new proof line is based on Appendix B of [7] and Polyak’s momentum proof. We describe our proof sketch here.
> 1. For Theorem 1: First, we prove that there is no false positive in $x^k$ for all $k$. Second, we prove that the support of $x^k$ is fixed as $k$ large enough. Finally, on the fixed support, the iteration formula (13) reduces to a linear iteration, and we apply the spectral analysis used in Polyak’s proof to obtain the convergence rate. The key change is that we calculate an upper bound of the recovery error caused by the noise and apply the upper bound in the first and third steps in the above proof sketch.
> 2. For Theorem 2: First we follow the same proof line with the first and second steps in the proof of Theorem 1. Then we apply the superlinear convergence of conjugate gradient on a symmetric linear system to obtain the convergence rate.
>
> Due to the space limitation of rebuttal, we will include the full proof in our final paper. The modified convergence rates are listed below:
>
> With the same assumption $|| \varepsilon ||_1 <= \sigma$ with that in [7], we have
>
> LISTA-CPSS [7]: \
> $||x^{(k)} - x^*||_2 \leq O(c^k) + C_1 \sigma$ (linear convergence with rate c)
>
> ALISTA-MM: In Theorem 1, the error bound is modified as: \
> $||x^{(k)} - x^*||_2 \leq O( c_1 * c_2 * c_3 … * c_k ) + C_2 \sigma,$\
> where $C_2 <= C_1$ and $c_k < c$ as k large enough. (faster linear convergence with rate $c_k$)
>
> ALISTA-MM with instance adaptive parameter: In Theorem 2, the error bound is modified as:\
> $||x^{(k)} - x^*||_2 \leq O( a_1 * a_2 * a_3 … * a_k ) + C_3 \sigma,$\
> where $C_3 <= C_1$ and $a_k \to 0$ as $k$ large enough. (superlinear convergence)
>
> ---------------------------------------------------------------------
>
> * Explanation of faster updates in curvature dimensions in Ln171.
>
> Thanks for the valuable comment. Our motivation to add the momentum is that acceleration techniques by momentum are well established in the optimization literature. An algorithm with the momentum term is usually faster than the algorithm without momentum. We will clarify L171 in our revision.
>
> To address the curiosity, a larger difference between $x^k$ and $x^{k-1}$ may not lead to a larger update of $x^{k+1}$ because we should consider not only the magnitude of $(x^k-x^{k-1})$ but also its direction.
>
> Take gradient descent on $\min_{x} F(x)$ as an example:
>
> Vanilla gradient descent: $x^{k+1} = x^k - \alpha  \nabla F(x^k)$,\
> where $\nabla F(x^k)$ means the gradient of $F$ at the point $x^k$.
>
> Adding a small momentum to gradient descent: \
> $x^{k+1} = x^k - \alpha  \nabla F(x^k) + \beta (x^k - x^{k-1}) $                              (1)\
> Since $\beta$ is small, we have\
> $x^k - x^{k-1} \approx  - \alpha  \nabla F(x^{k-1})$.\
> Then (1) can be approximately written as\
> $x^{k+1}  \approx  x^k - \alpha  \nabla F(x^k) - \beta  \alpha  \nabla F(x^{k-1})$
>
> In low curvatures,  the directions of $\nabla F(x^k)$ and $\nabla F(x^{k-1})$ are usually consistent, $x^{k+1}$ goes further along that direction.\
> In high curvatures, the directions of $\nabla F(x^k)$ and $\nabla F(x^{k-1})$ are not consistent, the update of $x^{k+1}$ may not have large magnitude.
>
> ----------------------------------------------------
>
> * Why choose the L0 norm of x^k in Ln214.
>
> From the perspective of motivation, using the L0 norm of $x^k$ as a proxy for weighting the momentum factor stems from the conclusion in eq. 17, where the momentum factor $\beta^{(k)}$ is proportional to “$s$” asymptotically, where $s$ is the sparsity of ground truth sparse vector $x^*$ in eq. 1. Therefore, l0 norm of $x^k$ is a natural choice to approximate $s$, considering that $x^k$ will gradually converge to $x^*$ and hence $||x^k||_0$ to $s$.
>
> ------------------------------
>
> * L179: What proofs the claim you already make in this line?
>
> The claim in L179 is proved by Theorem 1 that follows.
>
> --------------------------------
>
> * Why choose 16 layers in experiments?
>
> Firstly, 16 layers is a typical setting considered in previous works [7,14,25]. The reasoning behind this choice is that the traditional unrolling methods unroll the iterative algorithm into a feed-forward neural network with a **limited number** of layers, and train the model to have optimal performance within those limited layers. On the other hand, increasing the number of layers also brings difficulty in training the unrolled network, resulting in smaller gains from adding more layers.
>
> --------------------------------------
>
> * Perform grid-search directly on the unseen testing data.
>
> Thanks for the valuable comment. If the unseen data comes from a similar distribution with the training data, we don’t need to train the model again. If we repeatedly solve a certain type of problem, the training is a one-time cost.
>
> Our work is motivated by the unfolding methodology and therefore follows the standard machine learning settings where we have access to a set of training samples on which we will search for the best hyperparameters. The performance of different methods then will be evaluated on a hold-out set of testing samples to see if they “generalize” well to the testing samples.
>
> However, we agree with you that HyperLISTA can be seen as an algorithm instead of an unfolded neural network. It can be optimized directly on the unseen test data, as long as one wants to afford the cost of re-searching hyperparameters on each dataset although it is much lower than training using back-propagation.
>
> Hence, our approach stands at the intersection of machine learning (which emphasizes training on one sample set once and being directly applied latter), and hyperparameter tuning (which needs to be done on every new dataset)

---

> ### Comment · Area_Chair_bfqX · 2021-09-02
> **Please take a look at fXkT's review**
>
> Reviewer fXkT has cited a number of strengths and reasons for accepting the paper. Can you take a look at their review and author response and comment on your thoughts? Is there anything you disagree with or feel that they missed?

---

### Official Review · Reviewer_fXkT · 2021-07-11

**Rating:** 8
**Confidence:** 5

**Summary:**

This paper proposes an ultra-light variant of LISTA, called HyperLISTA, for solving sparse linear inverse problems. The authors’ adaptive parameterization reduces the training of HyperLISTA to only tuning three instance- and layer-invariant hyperparameters. HyperLISTA is theoretically proved and empirically observed to have super-linear convergence rates: the first time ever so in LISTA literature. It also leads to superior test-time adaptivity.

**Ethical Concerns:**

No ethic concerns

**Limitations And Societal Impact:**

Theoretical work. No concern here.

**Main Review:**

This paper adds a distinguished breakthrough to the LISTA literature, and I can see the work to become very influential in this booming community.

The theory part made substantial extensions from [14,5]. The seminal ALISTA [14] constructed the “optimal" model weights for achieving lower-bound linear rate, but that construction was not practically achievable due to requiring ground-truth sparse codes. The recent LSTM-based approach in [5] was theoretically motivated by revisiting the recovery guarantees of ALISTA and may achieve tighter error bounds. All previous results sit with linear rates for ALISTA.

This paper for the first time demonstrates ALISTA achieves a better convergence rate, and in particular is superlinearly convergent, with instance-optimal parameters. Although [5] had similar observations, this paper shows such instance-optimal parameters can be constructed in a way to have ALISTA achieve superlinear convergence, which is a significant step forward. Besides, Section 3.2 was also the first to unify the proof schemes in ALISTA and the momentum acceleration (Polyak’s heavy ball), which was not feasible previously.

The new theoretical results of Theorems 1 and 2 lead to a practical approach of automatically and adaptively calculating the parameters in a layer-progressive way. By that, HyperLISTA has reduced the complexity of learnable parameters to nearly the extreme: only three hyperparameters. That enables more efficient solving via cheaper gradient-free methods in HPO. For example, the authors showed even a grid search can do well, unlike previous LSTM black boxes.

Experiments find this compact adaptive paramterization can be self-adaptive to different sparsity levels, non-zero element magnitudes, additive noise levels, and even unrolled layer numbers. Overall, HyperLISTA achieves faster convergence on the seen distribution (i.e., the training data distribution), and shows better adaptivity on unseen distributions, while being ultra-lightweight.

Reading this paper has also been a joy. The writing is very good and easy to follow. Section 3 is well organized into a thoughtful and logical progression.

Overall I’m very positive about this work. The following suggestions are for the authors’ reference:
-	The re-parameterization of W in Section 3.1 was inspired by a very relevant prior work [1]. However, the authors never compared with [1] in their experiments (only NA-LISTA was compared). This one should be either explained or added.
-	Consider reporting ALISTA and other baselines too, in Figure 6’s unrolling iteration plot. Naïve options to extrapolate may include copying and re-using last-iteration hyperparameters or using some ad-hoc decay schedule. I guess that won’t hurt your main conclusion, just suggested for completeness.
-	[Nitpick] All experiments are on synthetic data. Following the standard of most LISTA papers nowadays, the authors shall consider reporting on natural image experiments too.
-	[Nitpick] Figure 5 shows a few test-time adaptivity settings. More settings may be considered, such as changing the non-zero element distribution family (currently, still Gaussian but altering std), or dictionary mismatch (like in robust ALISTA).


**Time Spent Reviewing:**

6 hourse

---

> ### Author Response · Authors · 2021-08-10
> **Response to Reviewer fXkT**
>
> Dear reviewer, thank you for your positive feedback and constructive suggestions for improving this work.
>
> **Comparison with Ada-LISTA** Thank you for pointing out the missing comparison with Ada-LISTA as an important baseline. During the rebuttal period, we follow the original implementation of Ada-LISTA and Ada-LFISTA (the accelerated version of Ada-LISTA with momentum) and train them in the noiseless setting (i.e., the setting used in Figure 5.c). Results show that 16-layer Ada-LISTA and Ada-LFISTA achieve -41.08dB and -45.55dB NMSE respectively at the last layer, which is much higher than the -90dB NMSE of HyperLISTA. We will include Ada-LISTA as a baseline in more settings in the final version.
>
> **Extrapolating ALISTA and other baselines** Thank you for your suggestion on extending more baselines to more iterations in Figure 6. We tried the extrapolation option on ALISTA by re-using the trained parameters in the last layer, finding that the resulting NMSE of ALISTA will be stuck around -72.5dB, which is much worse than HyperLISTA which easily achieves -130dB NMSE. We will include this and more baseline in Figure 6 in the revision.
>
> **Natural image experiments** Thank you for the suggestion! Although our work is more theory-oriented, we completely agree with your suggestion and think it will be interesting to apply HyperLISTA to real-world data. We are now implementing this and will update here as soon as we have the results.
>
> **Adaptivity to changes in non-zero distributions and dictionary** Thank you for the comments. We think these are interesting train-test mismatch settings and we are happy to include them in the final version when we have the results. For your curiosity here, we did try the setting of mismatched dictionaries, where the new dictionaries are independently sampled from the same Gaussian distribution. We find that HyperLISTA can adapt well in this simple case.

---

> > ### Comment · Reviewer_fXkT · 2021-08-17
> > **I am satisfied with the rebuttal.**
> >
> > The authors clarify my concerns. I will keep my score unchanged.

---

> > > ### Author Response · Authors · 2021-08-20
> > > **Thanks and An Update on the Natural Image Experiment**
> > >
> > > We thank the reviewer for the reply and we are glad that our response clarified the reviewer's concerns.
> > >
> > > We also followed the reviewer's suggestion and conducted a set of compressive sensing experiments on the BSD500 dataset, following a similar experiment setting in [7]. We compared HyperLISTA with FISTA, LISTA, ALISTA [14], and also Recon-Net (K. Kulkarni et al., 2016). All models are trained on 51,200 image patches collected on the BSD500 dataset and are tested on the Set 11 images used in Recon-Net. We report the average PSNR of the recovered images in the table below. More specific settings of the experiment can be found in our response to Reviewer WJ5N.
> > >
> > > | Method | FISTA - 16 iters | FISTA - 1,000 iters | LISTA | Recon-Net | ALISTA | HyperLISTA |
> > > |:------:|:----------------:|:-------------------:|:-----:|:---------:|:------:|:----------:|
> > > |  PSNR/dB  |       29.87      |        30.92        | 30.85 |   31.39   |  32.24 |    33.46   |

---

### Official Review · Reviewer_NZWV · 2021-07-15

**Rating:** 5
**Confidence:** 2

**Summary:**

This paper proposed a modification to the ALISTA algorithm which offers a better convergence rate.

**Main Review:**

This paper is out of my field and I would struggle to make informed comments about its substance. However I can state with confidence that this paper does not meet NeurIPS standards when it comes to presentation, clarity, phrasing and lack of typos. This is a non-exhaustive list of some of what I mean:

- You have an entire sentence duplicated (l43-44 vs l55-56). This confusion probably arose because the entire second half of your introduction should go into "related work".
- In line 57 you refer to the dictionary as D but it is introduced as A in equation 1, are you speaking of the same thing..?
- In the introduction you define "model-free" methods as "Deep learning base black-box models" which is confusing because that's not model-free by definition.
- trains -> training in first line of the abstract
- typo l39 "for" x2
- since it is a key concept, you should explain what unrolling means in the context of LISTA in the introduction.
- this sentence needs rephrasing: "Although such transformation is difficult to optimize, the authors argued that the data63 driven training process practically learned that, and hence the acceleration effect"
- typo "the" x2 in l85
- this sentence is confusing and needs rephrasing: "Throughout this paper, we refer “parameters” as learnable weights, thresholds theta(k), step sizes gamma(k) that are directed defined in LISTA models". What is "directed defined"? Is it another typo?
- typo in conclusion "trained more efficiently and at the same time more are more adaptive"


I would recommend proof reading your submission carefully to correct grammar/typos before resubmitting.

**Time Spent Reviewing:**

5

---

> ### Author Response · Authors · 2021-08-10
> **Response to Reviewer NZWV**
>
> Dear reviewer, thank you for the effort reviewing this work and listing the typos and inaccurate statements. We promise to fully proofread the whole manuscript and significantly improve the writing quality in the revision.
>
> **Duplicated sentences.** Sorry for the duplication. The first paragraph in the “related work” section is the expanded explanation with further details about the theoretical efforts for theoretical understanding of LISTA in the literature.
>
> **Inconsistent notation for dictionary.** Yes, we mean the same thing by the “dictionary”. It should be “dictionary A” in ln 57. We will fix this in the revised version.
>
> **Model-free definition.** Sorry for causing the confusion. Here, we use “model-free” to describe the deep architectures that are designed without considering the sparse inverse problem structure (formulated in eq. 1). In contrast, “model-based” refers to the deep architectures constructed by taking reference to the original sparse inverse problem. We agree that the terms here can be simplified and we will avoid using these two terms in the final version. However, we humbly believe that this has no impact on the story of this work.
>
> **The concept of “unrolling”.** Unrolling is the process of conversion that turns an iterative algorithm into a recurrent neural network or a feed-forward neural network with a specific number of layers. We would like to humbly point out that *unrolling* is a well-accepted concept and terminology used in the field, e.g., in the review paper “Algorithm unrolling: Interpretable, efficient deep learning for signal and image processing.” by V. Monga et al. We will cite this reference and clarify the concept of unrolling in the final version.
>
> **Rephrasing “Although such transformation is difficult to optimize, the authors argued that the data driven training process practically learned that, and hence the acceleration effect.”**
>
> This sentence is from ln62-63 in the “related work” subsection. We will rephrase it into:
> “Although it is difficult to search for the optimal transformation using classic optimization techniques, the authors argued that the data driven training process is practically able to learn that transformation from data, and hence the acceleration effect can be achieved.”
>
> **Rephrasing “Throughout this paper, we refer “parameters” as learnable weights, thresholds theta(k), step sizes gamma(k) that are directed defined in LISTA models...”**
>
> This is the first sentence in Section 2 (ln108). This sentence will be rephrased into:
> “Throughout this paper, we refer to “parameters” as the learnable weights in the models, such as the thresholds theta(k) and the step sizes gamma(k) that are defined in LISTA models…”
>
> **Other typos and grammatical errors.** Thank you for kindly pointing them out. We will fix them all in the final version.
>
> We also humbly refer you to our responses to Reviewer vByu, where we address many unclarity issues in the manuscript. We believe that the comments from you and Reviewer vByu helped very much for us to significantly improve the writing quality and readability, which hopefully makes our paper well over the standard of NeurIPS, especially considering our original substantial contributions made by our work as appreciated by Reviewer WJ5N and Reviewer fXkT.

---

> ### Author Response · Authors · 2021-08-26
> **Did we address your questions and concerns? Your feedback would be appreciated**
>
> Dear Reviewer NZWV,
>
> We would like to thank you for your valuable time and efforts pointing out many parts in the manuscript that are not clear or presented well. This helped us to improve its clarity and readability. As the discussing period is approaching its end, we wonder if our response addresses your questions and concerns. We would appreciate it if you could also kindly have a look at the rolling discussion between Reviewer vByu and us, where we performed and committed various efforts to improve presentation clarity . We promise to integrate everything in the rebuttal into the final manuscript.
>
> Thank you again for your time! It would be great if you could share more thoughts on our previous reply, and let us know if there is any additional question or concern.
>
> Best,
>
> Paper8022 Authors

---

> > ### Comment · Reviewer_NZWV · 2021-08-27
> > **reply**
> >
> > Unfortunately it's hard for reviewers to take your word for it when there are so many changes to be made. The list of typos/clarity issues I gave you wasn't exhaustive, and I still believe this paper should be resubmitted once appropriate effort has been put into presentation/clarity.

---

### Official Review · Reviewer_WJ5N · 2021-07-18

**Rating:** 6
**Confidence:** 5

**Summary:**

The paper is based on a branch of works named LISTA(Learned Iterative Shrinkage-Thresholding Algorithm), reducing hyperparameters to 3, while maintaining almost the same performance on seen data distributions. Theoritical explainations and experiments are also carried out.

**Limitations And Societal Impact:**

The authors have not discussed limitations of their work. By reducing the hyperparameters to 3, there might be some drawbacks comparing to original LISTA works. It is recommended to discuss if there are any drawbacks\limitations or not, and why.
It is not likely for the work to have any potential negative societal impact.

**Main Review:**

1. As is stated in line 9-10, the work of this paper trims down LISTA complexity to 3 hyperparameters. Considering the performance showed in figure 6 by experiments, the performance under certain circumstance is good, HyperLISTA outperforms NA-ALISTA. So the trimming-down can be a contribution.
But does the improvement come with drawbacks? This is not fully discussed in the paper. There is one major problem. Will the flexibility be influenced when reduce the number of hyperparameters to 3? If the flexibility will be influenced, what impact will it produce?

2. Typos.
In line 317 "more are more adaptive".

Plus, should the y-axis label "NMES" be "NMSE" in the figures?

3. In line 258 "We use synthesized datasets of 51,200 samples for training, 2,048 validation and 2,048 testing". As the paper claims that the work improves and extends the branch of works LISTA as is stated in line 9-10. To show that the proposed HyperLISTA is as widely applicable as former algorithms, more experiments on real-world datasets are suggested.

4. Extends question 1, it is suggested to introduce the real-world applications of LISTA, and how will the proposed HyperLISTA help in those applications.

**Time Spent Reviewing:**

3h

---

> ### Author Response · Authors · 2021-08-10
> **Response to Reviewer WJ5N**
>
> Dear reviewer, thank you for your review and constructive comments.
>
> **Contribution and limitation of HyperLISTA.** LISTA [12] is the first work to solve mathematical optimization problems by machine learning. It parameterizes an iterative optimization algorithm and unrolls this algorithm to a feedforward neural network that can be end-to-end trained.
>
> ALISTA [14] makes seminal contributions to this line of work. It established solid theories and constructed analytic weights that can replace the parameter weights in LISTA. Such results partially opened the black-box of the LISTA neural network and trimmed down the model from O(K N^2) to 2K parameters, where K is the number of layers, N is the number of columns of matrix A.
>
> However, the generalization ability is an issue for ALISTA. To address this issue, we propose HyperLISTA that calculates instance-adaptive parameters. The new parameterization brings three benefits:
> (1) The model can generalize to unseen data that follow different distributions with the training data (different sparsities, different noise levels, different magnitudes);
> (2) It trims the model complexity to only three hyperparameters;
> (3) Surprisingly, the convergence rate can be improved from linear to superlinear because the parameters are no longer tied over the instances.
> To the best of our knowledge, these points are the first time to unrolling literature.
>
> Since HyperLISTA is solidly grounded with theoretical contributions, the flexibility will NOT be influenced when the number of hyperparameters is reduced to 3. On the contrary, the flexibility is actually improved. We are able to use our trained model on unseen data with different sparsities, different noise levels, and different magnitudes, which are not feasible to ALISTA.
>
> In our problem settings, as long as Assumption 1 is satisfied, HyperLISTA converges faster without compromise. Beyond our problem settings, for example, a dictionary with perturbation is not considered now. But we could follow the “robust ALISTA” method to solve this problem in the future.
>
> **Experiments on real-world datasets.** Thank you for your suggestion on adding experiments on real-world datasets. While this work is more theory-driven, we are happy to apply HyperLISTA to real-world data. We are currently working on the implementation of this, and will update here as soon as we have the results.
>
> **Typos.** Thank you for pointing out the typos and the wrong axis labels. We will fix them in the revision and improve the overall writing quality.

---

> > ### Author Response · Authors · 2021-08-20
> > **Update of Experiments on Real-world Datasets**
> >
> > Dear Reviewer WJ5N,
> >
> > We would like to thank you again for your suggestion on adding experiments on real-world datasets. Following the suggestion, we conducted a set of compressive sensing experiments on natural images, following a similar setting in [7]. We randomly extract 16x16 patches (flatten into 256-dim vectors then) from the BSD500 dataset to form a training set of 51,200 samples, a validation and a test set of 2,048 samples respectively.
> >
> > We compressively measure the image signals (256-dim vectors) using a random Gaussian matrix $\Phi$ of shape 128x256. We do not add measuring noises. We apply a dictionary learning algorithm in (Y. Xu & W. Yin, 2014) to the training patches of BSD to learn a matrix $D$ of shape 256x512, where we assume any patch in a natural image can be represented by a sparse combination of the columns in $D$. We then use $A=\Phi D$ as the dictionary $A$ in eqn. 1. In summary, we denote the image patch as $f$. The measurement we observe, which is also the input to the network, is generated by $b = \Phi f$. We assume that there exists a sparse vector $x^*$ that satisfies $f \approx Dx^*$ and hence $b\approx Ax^* = \Phi D x^*$. We apply different methods to recover $x^*$ from measurement $b$ and then reconstruct the image patch by multiplying the recovered sparse vector with $D$.
> >
> > Because we do not have access to the underlying ground truth $x^*$ in this real-world scenario, we instead use the LASSO objective function as the loss function for training (of LISTA/ALISTA) or searching (of HyperLISTA), i.e.,
> >
> > $$ Loss = \mathbb{E}\left[\frac{1}{2} \|\|b - A \hat{x}(b)\|\|_2^2 + \lambda \|\|\hat{x}(b)\|\|_1\right], $$
> >
> > where $\hat{x}(b)$ is the sparse vector recovered from the measurement $b$ and $\lambda$ is the sparsity regularization coefficient. We set $\lambda=0.05$ in the experiment. We then reconstruct the image signal by $\hat{f}=D\hat{x}$.
> >
> > We train LISTA and ALISTA [14], and perform grid search to find optimal hyperparameters in HyperLISTA. These three models all have 16 layers. The model performance is evaluated by the average PSNR of the reconstructed images on the standard testing images in Set 11 used in Recon-Net (K. Kulkarni et al., 2016). The results are shown in the table below, where we also compare with FISTA and Recon-Net. We can see that HyperLISTA outperforms other baselines by clear margins. Note that the performance of LISTA is much lower than that reported in [7] where 4,000,000 patches are used for training, while we only use 51,200 training patches.
> >
> > | Method | FISTA - 16 iters | FISTA - 1,000 iters | LISTA | Recon-Net | ALISTA | HyperLISTA |
> > |:------:|:----------------:|:-------------------:|:-----:|:---------:|:------:|:----------:|
> > |  PSNR/dB  |       29.87      |        30.92        | 30.85 |   31.39   |  32.24 |    33.46   |
> >
> > We hope that this experiment could help to relieve your concern about the applicability of HyperLISTA in real-world scenarios and better support your positive evaluation of this work.
> >
> >
> > ### References
> >
> > Xu, Yangyang, and Wotao Yin. "A fast patch-dictionary method for whole image recovery." arXiv preprint arXiv:1408.3740 (2014).
> >
> > Kulkarni, Kuldeep, et al. "Reconnet: Non-iterative reconstruction of images from compressively sensed measurements." Proceedings of the IEEE Conference on Computer Vision and Pattern Recognition. 2016.

---

### Author Response · Authors · 2021-08-23
**To all reviewers**

We really appreciate all the reviewers for their valuable suggestions.
- We thank reviewer fXkT for strongly supporting the originality and significance of our paper.

- We thank reviewer vByu for giving detailed comments and being satisfied with our answers. We would appreciate it even more, if reviewer vByu could take the valuable time to provide us the way to clear his/her concerns on clarity and adjust the score. We provide two options in our rebuttal and are looking forward to hearing from reviewer vByu.

- We sincerely hope to have further discussion with reviewers WJ5N and NZWV to see if our response solves their concerns. We are confident that our response should have cleared the air, and we can clarify more if there is more need. We are happy to answer any additional questions and provide more information.

---

### Author Response · Authors · 2021-08-31
**A revised pdf is now available**

Dear reviewers and AC:

A revised pdf is now available in the following anonymous link: https://tinyurl.com/5xbm9hkr

- We did not immediately upload the pdf in the past few days, only because there were some delayed responses during our rolling discussion with reviewer vByu and AC, that made us think a PDF was no longer needed. The draft was actually ready before the weekend, and we happily provide it now.

- A proper amount of revisions and modifications have been integrated in the current version, as highlighted in the main text. Most of them are minor writing issues (e.g. clarifying notations, adding outline overview, explaining literature background, etc.), and have been fixed or polished easily. The main technical idea stays all the same.

- We thank all suggestions that help us improve writing and clarify. We also politely point out that our technical significance and novelty are solid and strong, as unanimously acknowledged.

- We would appreciate it if reviewers vByu and NZWV could please take a look and finalize their assessments on our work, hopefully more positively. We trust the reviewer and AC discussion would eventually lead to an informed and fair decision, and we thank everyone again for the valued efforts!

Best,

Paper8022 Authors

---

### Decision · Program_Chairs · 2021-09-27

**Decision:**

Accept (Poster)

**Comment:**

This work makes significant contributions to research into algorithm unrolling, by building upon the LISTA and ALISTA works. After providing a new ALISTA parameterization, and introducing a momentum term the authors both significantly reduce the complexity of tuning relative to prior work, and achieve significant performance gains. Most reviewer criticism centered on quality of writing, which was addressed in the rebuttal. Regarding the technical merit of the work, some reviewers offered extremely high praise arguing strongly for acceptance.